# Architecture and structural dynamics of the heteromeric GluK2/K5 kainate receptor

Nandish Khanra[1], Patricia MGE Brown[2†], Amanda M Perozzo[2], Derek Bowie[2], Joel R Meyerson[1]*

[1]Department of Physiology and Biophysics, Weill Cornell Medical College, New York, United States; [2]Department of Pharmacology and Therapeutics, McGill University, Montréal, Canada

**Abstract** Kainate receptors (KARs) are L-glutamate-gated ion channels that regulate synaptic transmission and modulate neuronal circuits. KARs have strict assembly rules and primarily function as heteromeric receptors in the brain. A longstanding question is how KAR heteromer subunits organize and coordinate together to fulfill their signature physiological roles. Here we report structures of the GluK2/GluK5 heteromer in apo, antagonist-bound, and desensitized states. The receptor assembles with two copies of each subunit, ligand binding domains arranged as two heterodimers and GluK5 subunits proximal to the channel. Strikingly, during desensitization, GluK2, but not GluK5, subunits undergo major structural rearrangements to facilitate channel closure. We show how the large conformational differences between antagonist-bound and desensitized states are mediated by the linkers connecting the pore helices to the ligand binding domains. This work presents the first KAR heteromer structure, reveals how its subunits are organized, and resolves how the heteromer can accommodate functionally distinct closed channel structures.

*For correspondence:
jrm2008@med.cornell.edu

Present address: † MRC Laboratory of Molecular Biology, Cambridge, United Kingdom

Competing interests: The authors declare that no competing interests exist.

## Introduction

Ionotropic glutamate receptors (iGluRs) respond to the neurotransmitter L-glutamate (L-Glu) to mediate the majority of fast excitatory synaptic transmission in the human brain (*Dingledine et al., 1999*). This central role has implicated iGluRs in major pathological brain conditions including depression, schizophrenia, stroke, Alzheimer's and Parkinson's disease (*Bowie, 2008*). The iGluR family has three subtypes termed kainate receptors (KARs), α-amino-3-hydroxy-5-methyl-4-isoxazole-propionic acid receptors (AMPARs), and *N*-methyl-D-aspartate receptors (NMDARs) (*Traynelis et al., 2010*). Each subtype possesses unique functional properties and fulfills distinct physiological roles (*Dingledine et al., 1999*; *Traynelis et al., 2010*). KARs are expressed throughout the central nervous system and act post-synaptically in depolarization and modulation of membrane excitability, and pre-synaptically to regulate excitatory and inhibitory neurotransmitter release (*Contractor et al., 2011*; *Lerma and Marques, 2013*). Their role in pain perception, epilepsy, and mood disorders has also made KARs important targets of small-molecule antagonist and modulator development (*Contractor et al., 2011*; *Larsen et al., 2017*; *Møllerud et al., 2017*).

KARs are tetramers that assemble with four identical subunits (homomers) or mixtures of different subunits (heteromers). Each subunit has a modular three-layer design with an amino terminal domain (ATD), ligand binding domain (LBD), and transmembrane domain (TMD) which forms part of a common pore (*Traynelis et al., 2010*). The tetramers have variable symmetry among their three domain layers (*Meyerson et al., 2016*), a feature that is also found in AMPARs and NMDARs (*Zhu and Gouaux, 2017*). KAR tetramers are generated from a pool of five subunits. GluK1, GluK2, and GluK3 subunits can form functional homomers (*Cui and Mayer, 1999*). GluK4 and GluK5 cannot form

functional channels on their own and are obliged to co-assemble with GluK1, GluK2, or GluK3 (*Herb et al., 1992*; *Werner et al., 1991*). GluK4-5 also have higher L-Glu affinity (*Herb et al., 1992*) and thus a lower activation threshold than GluK1-3 (*Barberis et al., 2008*; *Fisher and Mott, 2011*; *Mott et al., 2010*). Heteromers containing GluK5 subunits show enhanced permeation to polyamine pore blockers (*Brown et al., 2016*), distinct pharmacological properties (*Herb et al., 1992*; *Swanson et al., 2002*), and are endowed with slower deactivation kinetics (*Barberis et al., 2008*) which is proposed as central to their role in integrating synaptic signals (*Frerking and Ohliger-Frerking, 2002*; *Lerma and Marques, 2013*). Consequently homomers and heteromers occupy different functional niches in the brain, with heteromers thought to be the major KAR class (*Herb et al., 1992*; *Petralia et al., 1994*). Receptors containing either GluK4 or GluK5 are also specifically linked to diseases such as depression (*Catches et al., 2012*), epilepsy (*Das et al., 2012*), autism (*Aller et al., 2015*), and schizophrenia (*Greenwood et al., 2016*). Despite their significance, no KAR heteromer structure has been reported, which presents a major knowledge gap made more salient by recently reported structures of NMDAR (*Karakas and Furukawa, 2014*; *Lee et al., 2014*; *Lü et al., 2017*) and AMPAR heteromers (*Herguedas et al., 2019*; *Zhao et al., 2019*).

KARs are ligand-gated cation channels which respond to L-Glu released into the synapse. L-Glu binding triggers the receptor to enter an active state with an open channel which allows cations to cross the membrane. The channel closes again if L-Glu is rapidly removed (deactivation) or if L-Glu exposure is sustained (desensitization) (*Barberis et al., 2008*). Recent structures have shown how GluK2 and GluK3 homomers are organized in the absence of L-Glu while bound to competitive antagonists and after the receptors have desensitized (*Kumari et al., 2019*; *Meyerson et al., 2014a*; *Meyerson et al., 2016*; *Schauder et al., 2013*). The desensitized structures revealed that KAR homomers feature a 'desensitization ring' motif which is proposed to facilitate channel closure and account for their slow recovery from desensitization (*Meyerson et al., 2016*). To date this motif has not been observed in AMPAR or NMDAR structures, raising the question of whether it is unique to KAR homomers or if KAR heteromers employ a similar desensitized structure.

Here, we report structures of the GluK2/GluK5 heteromer (GluK2/K5), proposed to be the major KAR in the brain (*Herb et al., 1992*; *Petralia et al., 1994*), in resting (apo), antagonist-bound (6-cyano-7-nitroquinoxaline-2,3-dione [CNQX]), and agonist-bound (L-Glu) desensitized states. The apo and antagonist-bound structures allow us to address questions of KAR heteromer architecture, symmetry, and the organization of its three structural 'layers'. With the L-Glu-bound structure, we reveal the significant structural changes which accompany heteromer desensitization and compare the structure to the desensitized GluK2 homomer. We conclude with structural analysis to answer how antagonist-bound and desensitized GluK2/K5 states can both maintain a closed cation channel despite such different LBD arrangements. Critically, this work provides an expanded foundation for understanding differences and similarities among KAR (*Kumari et al., 2019*; *Meyerson et al., 2014a*; *Meyerson et al., 2016*), AMPAR (*Herguedas et al., 2019*; *Zhao et al., 2019*), NMDAR (*Karakas and Furukawa, 2014*; *Lee et al., 2014*; *Lü et al., 2017*), and recently characterized ionotropic orphan delta (GluD) receptor structures (*Burada et al., 2020a*; *Burada et al., 2020b*).

## Results

### Receptor isolation, functional testing, and structure determination

In order to obtain a structure of GluK2/K5, it was necessary to generate receptor protein of sufficient quality and quantity for cryo-electron microscopy (cryo-EM) imaging. However, the GluK2/K5 receptor has low expression levels in recombinant systems (*Hayes et al., 2003*; *Nasu-Nishimura et al., 2006*; *Reiner et al., 2012*; *Ren et al., 2003*). To surmount this problem, we developed optimized protein co-expression constructs for GluK2 and GluK5. In both subunits we mutated select cysteines in the TMD, and for GluK5 we also removed the flexible C-terminus which contains multiple endoplasmic reticulum retention signals (*Nasu-Nishimura et al., 2006*; *Ren et al., 2003*) (see 'Materials and methods' for details). We refer to these constructs as GluK2$_{em}$ and GluK5$_{em}$ (*Figure 1—figure supplement 1*, *Figure 1—figure supplement 2A*).

We next compared the functional properties of wild-type and cryo-EM constructs in order to validate that GluK2$_{em}$ and GluK5$_{em}$ form functional receptors (*Figure 1—figure supplement 2* and *Table 1*). For these experiments, we used outside-out patch clamp electrophysiology to measure

channel current decays in response to a prolonged (250 ms) exposure to L-Glu (i.e. desensitization) or to a brief (1 ms) pulse of L-Glu (i.e. deactivation). Recordings of cells transfected with wild-type GluK2 or co-transfected with wild-type GluK2 and GluK5 show rapid desensitization (*Figure 1—figure supplement 2B,D*), as expected (*Barberis et al., 2008*). In addition, the two conditions show divergent responses to short L-Glu exposure with GluK2 deactivating quickly and the combination of GluK2 and GluK5 deactivating slowly (*Figure 1—figure supplement 2C,D*). The slow channel closure in the heteromer occurs because GluK5 subunits have high affinity for L-Glu, and occupancy of their LBDs is sufficient to sustain channel activation after L-Glu application is halted (*Barberis et al., 2008*). Importantly, co-expression of GluK2 and GluK5 subunits gives a mixture of GluK2 homomers and GluK2/K5 heteromers (as noted, GluK5 alone does not form functional homomers), and measuring deactivation provides a way to isolate a characteristic heteromer signal (*Barberis et al., 2008*). We tested GluK2$_{em}$ alone and found it shows the expected rapid desensitization from prolonged L-Glu treatment and rapid deactivation after a short L-Glu pulse (*Figure 1—figure supplement 2E*). This confirmed GluK2$_{em}$ can form functional homomers with similar kinetic properties to wild-type GluK2 (*Table 1*). We next co-expressed GluK2$_{em}$ and GluK5$_{em}$ and observed rapid desensitization from a mixture of GluK2$_{em}$ homomers and GluK2$_{em}$/GluK5$_{em}$ heteromers (referred to as GluK2/K5$_{em}$ heteromers) (*Figure 1—figure supplement 2F*). In turn, the deactivation current decays slowly, indicating the inclusion of the GluK5$_{em}$ subunits and formation of functional GluK2/K5$_{em}$ heteromers (*Figure 1—figure supplement 2F*). Taken together, GluK2/K5$_{em}$ heteromers exhibited the expected functional properties of rapid desensitization and relatively slow deactivation with kinetic parameters comparable to wild-type receptors (*Table 1*). We concluded that GluK2$_{em}$ and GluK5$_{em}$ can form functional heteromers, and the two expression constructs were used to produce GluK2/K5$_{em}$ protein using the BacMam method (*Goehring et al., 2014*; *Morales-Perez et al., 2016*) (see 'Materials and methods' and *Figure 1—figure supplement 3*).

We first pursued a structure of GluK2/K5$_{em}$ in an apo resting state and imaged the receptor by cryo-EM. The data were processed using single particle analysis without symmetry applied and a structure of GluK2/K5$_{em}$-apo was resolved to 7.5 Å global resolution (*Figure 1—figure supplement 4*, *Table 2*). At the calculated resolution, the structure is suitable to observe domain positions and we unambiguously identified ATDs, LBDs, and TMDs. The four LBDs arrange as two pairs of dimers (*Figure 1—figure supplement 4*) which accord with a canonical model of a resting state structure (*Dürr et al., 2014*). No apo state KAR LBD structures were available to analyze the status of the LBDs, but fitting a crystal structure of the apo GluA2 AMPAR LBD (PDB: 1FTO) into the cryo-EM densities confirmed that all four LBDs are compatible with an apo conformation (*Figure 1—figure supplement 5A*). Lastly, the ATD layer has a subtle 'tilt' away from the receptor central axis and the linkers connecting the ATD and LBD layers were not resolved (*Figure 1—figure supplement 4*). Taken together these features may indicate some degree of relative mobility between the ATD and the LBD-TMD region. We note that this tilt is similar to ATD tilts observed in AMPAR structures solved by X-ray crystallography (*Sobolevsky et al., 2009*; *Yelshanskaya et al., 2014*) and cryo-EM

**Table 1.** Summary of kainate receptor (KAR) decay kinetics for wild-type and cryo-electron microscopy (cryo-EM) constructs.

| Receptor | Desensitization kinetics | | | | | | | |
| --- | --- | --- | --- | --- | --- | --- | --- | --- |
| | $\tau_1 \pm$ SEM (ms) | % | $\tau_2 \pm$ SEM (ms) | % | $\tau_3 \pm$ SEM (ms) | % | Weighted $\tau$ (ms) | n |
| GluK2 | 6.3 ± 0.3 | 92 | 46 ± 8 | 8 | – | – | 7.5 | 27 |
| GluK2em | 7.7 ± 0.7 | 95 | 41 ± 8 | 5 | – | – | 8.8 | 7 |
| GluK2/K5 | 2.8 ± 0.1 | 92 | 31 ± 6 | 8 | – | – | 4.2 | 26 |
| GluK2/K5em | 2.4 ± 0.1 | 96 | 20 ± 4 | 4 | – | – | 2.9 | 6 |
| | Deactivation kinetics | | | | | | | |
| Receptor | $\tau_1 \pm$ SEM (ms) | % | $\tau_2 \pm$ SEM (ms) | % | $\tau_3 \pm$ SEM (ms) | % | Weighted $\tau$ (ms) | n |
| GluK2 | 3.3 ± 0.2 | 97 | 21 ± 3 | 3 | – | – | 3.9 | 28 |
| GluK2em | 4.0 ± 0.4 | 98 | 16 ± 3 | 2 | – | – | 4.2 | 9 |
| GluK2/K5 | 1.2 ± 0.1 | 44 | 10 ± 1 | 21 | 60 ± 2 | 35 | 23.6 | 35 |
| GluK2/K5em | 1.3 ± 0.2 | 60 | 15 ± 5 | 18 | 81 ± 16 | 22 | 18.5 | 6 |

**Table 2.** Cryo-electron microscopy (cryo-EM) data collection and processing.

| | GluK2/K5-apo | GluK2/K5-CNQX (full-length) | GluK2/K5-CNQX (ATD) | GluK2/K5-CNQX (LBD-TMD) | GluK2/K5-L-Glu (full-length) | GluK2/K5-L-Glu (ATD) | GluK2/K5-L-Glu (LBD-TMD) |
|---|---|---|---|---|---|---|---|
| Magnification | 81,000 | 36,000 | 36,000 | 36,000 | 36,000 | 36,000 | 36,000 |
| Voltage (kV) | 300 | 200 | 200 | 200 | 200 | 200 | 200 |
| Electron exposure ($e^-/Å^2$) | 51.23 | 50–53 | 50–53 | 50–53 | 50–53 | 50–53 | 50–53 |
| Defocus (μm) | 1.6 | 0.4–4.8 | 0.4–4.8 | 0.4–4.8 | 0.4–4.8 | 0.4–4.8 | 0.4–4.8 |
| Pixel size (Å) | 1.083 | 1.096 | 1.096 | 1.096 | 1.096 | 1.096 | 1.096 |
| Symmetry imposed | C1 | C1 | C1 | C2 | C1 | C1 | C2 |
| Initial particle images (#) | 1,778,627 | 51,898,826 | 51,898,826 | 51,898,826 | 28,586,529 | 28,586,529 | 28,586,529 |
| Final particle images (#) | 90,027 | 1,021,916 | 540,580 | 184,945 | 573,403 | 241,849 | 140,028 |
| Map resolution (Å) | 7.5 | 5.3 | 3.6 | 4.2 | 5.8 | 3.8 | 4.3 |
| Fourier shell correlation (FSC) threshold | 0.143 | 0.143 | 0.143 | 0.143 | 0.143 | 0.143 | 0.143 |
| Map resolution range (Å) | 6.5–8.0 | 4.0–6.0 | 3.5–4.0 | 4.0–5.0 | 4.0–6.0 | 3.5–4.0 | 4.0–5.0 |

(*Nakagawa, 2019*) and may reflect a general structural requirement for flexible ATD-LBD linkers to accommodate LBD movements during receptor gating.

We noted that many iGluR structures have been solved in the presence of saturating concentration of antagonist to stabilize the receptor assembly (*Chou et al., 2020*; *Meyerson et al., 2016*; *Nakagawa, 2019*; *Sobolevsky et al., 2009*). To maximize our chances of obtaining a high-resolution structure, we adopted this approach and selected the antagonist CNQX because it binds with low micromolar affinity to GluK2/K5 heteromers (*Alt et al., 2004*). We prepared samples of GluK2/K5em with 1 mM CNQX, imaged with cryo-EM, processed the data using single particle analysis without symmetry applied, and refined a structure to 5.3 Å resolution (*Figure 1—figure supplements 6 and 7*). The intermediate resolution motivated us to use particle subtraction in Relion (*Zivanov et al., 2018*) to independently resolve the ATD layer and the LBD-TMD assembly. After subtracting the LBD and TMD layers from the particle set, we refined the ATD layer to 3.6 Å resolution, and then did the opposite by subtracting the ATD layer and refined the LBD-TMD assembly to 4.2 Å (*Figure 1—figure supplements 6 and 7*). The two independent structures were used to make a full-length model which was used for analysis (*Figure 1*).

## Receptor organization and symmetry

The GluK2/K5em-CNQX quaternary architecture has three layers with ATDs at the 'top', the LBDs in the 'middle', and the TMD at the 'base' (*Figure 1A–F*). The overall profile of the receptor resembles the apo state with two pairs of LBD dimers and slightly tilted ATD (*Figure 1B, C and E*, *Figure 1—figure supplement 4*). Like many iGluR structures solved by cryo-EM, the resolution of the density map is not sufficient to determine the presence or absence of ligands (*Jalali-Yazdi et al., 2018*; *Kumari et al., 2019*; *Lü et al., 2017*; *Tajima et al., 2016*; *Twomey et al., 2016*; *Zhao et al., 2016*), and rather ligand binding is inferred from the sample conditions and protein structure. To test if the LBDs are compatible with antagonist occupancy, we measured their similarity to an antagonist-bound GluK2 LBD crystal structure (PDB: 5CMK) and also compared the LBDs to an L-Glu-bound crystal structure (PDB: 1S50) (*Figure 1—figure supplement 5B*). Inspection of the conformational differences and the measured RMSD values showed clearly that the LBDs had 'open' binding clefts that were similar to the antagonist-bound LBD reference structure.

We next identified the GluK2em and GluK5em subunits in the assembly by exploiting successful visualization of the mutually exclusive N-linked glycosylation sites on GluK2 and GluK5 subunits (*Figure 1—figure supplement 8*). This shows that GluK2em subunits form an interface between the two halves of the ATD layer (B/D positions), while the GluK5em ATDs heterodimerize with GluK2em ATDs which position them at the periphery of the ATD layer (A/C positions) (*Figure 1D*, *Figure 1—figure supplement 8*). Assigning the ATD identities allowed us to conclude that GluK2/K5 receptors

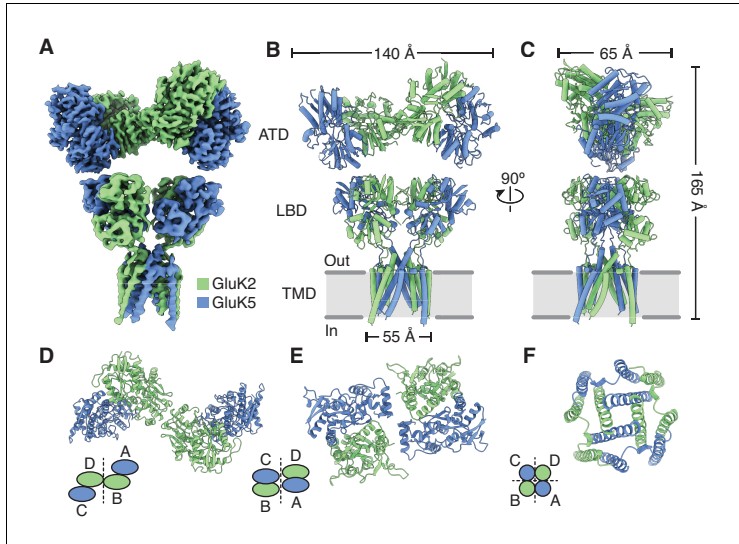

**Figure 1.** Structure of the GluK2/K5 heteromer. (**A**) Cryo-electron microscopy (cryo-EM) structure of the GluK2/K5$_{em}$ heteromer in a 6-cyano-7-nitroquinoxaline-2,3-dione (CNQX)-bound state with GluK2$_{em}$ and GluK5$_{em}$ subunits rendered in green and blue, respectively. The panel shows the map for the amino terminal domain (ATD) layer and the map for the ligand binding domain (LBD)-transmembrane domain (TMD) assembly which were reconstructed independently, as described in the text. (**B and C**) Molecular model for the receptor colored as in (**A**) and shown from two different views parallel to the membrane. (**D–F**) The three layers of the GluK2/K5$_{em}$ heteromer as viewed from the extracellular space. The local symmetries of the ATD (D, twofold), LBD (E, twofold), and TMD (F, fourfold) are illustrated.

The online version of this article includes the following figure supplement(s) for figure 1:

**Figure supplement 1.** Heteromer construct design and structural annotation.

**Figure supplement 2.** Heteromer construct illustrations and functional validation.

**Figure supplement 3.** Heteromer biochemistry and cryo-electron microscopy (cryo-EM).

**Figure supplement 4.** Cryo-electron microscopy (cryo-EM) image processing and structure for GluK2/K5$_{em}$ in an apo resting state.

**Figure supplement 5.** Analysis of ligand binding domain (LBD) conformations.

**Figure supplement 6.** Cryo-electron microscopy (cryo-EM) image processing workflow for GluK2/K5$_{em}$ with 6-cyano-7-nitroquinoxaline-2,3-dione (CNQX).

**Figure supplement 7.** Resolution determination for GluK2/K5$_{em}$ structures with 6-cyano-7-nitroquinoxaline-2,3-dione (CNQX).

**Figure supplement 8.** Mutually exclusive sites of glycosylation on GluK2 and GluK5 subunits.

---

assemble with 2:2 stoichiometry which is consistent with single molecule experiments (*Litwin et al., 2020*; *Reiner et al., 2012*), molecular dynamics simulation (*Paramo et al., 2017*), and an X-ray structure of isolated GluK2/K5 ATDs (PDB: 3QLV) (*Kumar et al., 2011*).

The early studies on GluK4 and GluK5 showed that the subunits are unable to form functional receptors on their own and must co-assemble with GluK1, GluK2, or GluK3 (*Herb et al., 1992*; *Werner et al., 1991*). Recent work on the GluK4 LBD (*Kristensen et al., 2016*) and molecular dynamics simulations on GluK2/K5 (*Paramo et al., 2017*) suggest that GluK4 and GluK5 LBDs function as heterodimers with the other three KAR subunits. Based on these results we hypothesized that in the full-length receptor, GluK2 and GluK5 LBDs co-assemble as two pairs of heterodimers rather than as two pairs of homodimers. Analysis of the GluK2/K5$_{em}$-CNQX structure validates this hypothesis and shows that GluK2/K5 LBD heterodimers form with GluK5 in pore-proximal (A/C) positions and GluK2 in pore-distal (B/D) positions (*Figure 1—figure supplement 8*). We next compared the LBD layer of GluK2/K5$_{em}$-CNQX to LBD layers of an antagonist-bound KAR homomer (GluK2) (*Meyerson et al., 2016*), AMPAR (GluA1/A2) (*Herguedas et al., 2019*), and NMDAR heteromers (GluN1b/N2B) (*Chou et al., 2020*; *Figure 2*). The profile of the GluK2/K5 LBD layer is most similar to GluK2 and the AMPAR when viewed from the extracellular space (*Figure 2A*). This observation is reinforced when considering subunit orientation by using helices B and G as references across the

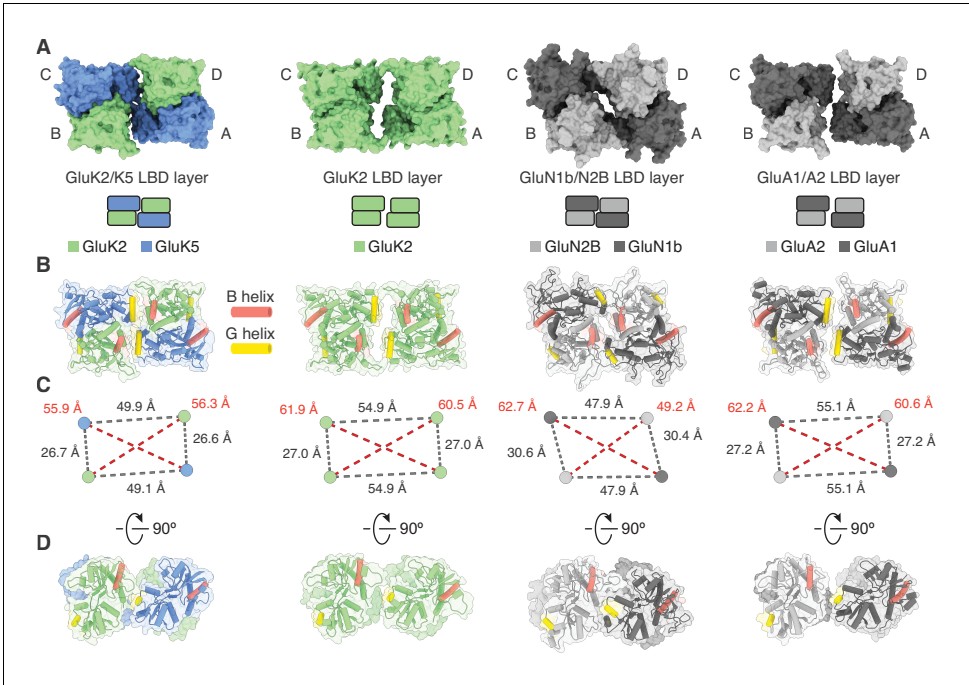

**Figure 2.** Analysis of the GluK2/K5 ligand binding domain (LBD) layer. (A) Extracellular view of LBD layers for (left to right) GluK2/K5em, GluK2, GluN1b/N2B, and GluA1/A2. The LBD layers are extracted from full-length antagonist-bound cryo-electron microscopy (cryo-EM) structures: GluK2/K5em-CNQX, GluK2-LY466195 (PDB: 5KUH), GluN1b/N2B-SDZ 220–040/L689,560 (PDB: 6WHU), GluA1/A2-γ8-NBQX (PDB: 6QKC). (B) LBD layers as viewed in (A), but shown with secondary structure and helices B (red) and G (yellow) colored as visual guides for subunit orientation. (C) Intersubunit measurements for heteromeric LBD layers as shown in (A). Measurements are between the centers of mass for each LBD as calculated in UCSF ChimeraX. (D) LBDs are shown colored as in (B) but viewed parallel to the membrane.

three LBD layers (*Figure 2B,D*). We then considered the positioning of each LBD by calculating distances between their centers of mass in UCSF ChimeraX (*Goddard et al., 2018*). Measurements between LBDs *within* a dimer were similar for all the structures and ranged between 26.7 and 30.6 Å (*Figure 2B*). When comparing *between* LBD dimers, it is clear that GluK2/K5em and GluN1b/N2B form an overall more compact arrangement with dimer-dimer distances of 49.9 and 47.9 Å, respectively. This contrasts with distances of 54.9 and 55.1 Å for GluK2 and GluA1/A2 (*Figure 2B*). From these measurements we conclude that the GluK2/K5 LBD layer is more compact than the GluK2 homomer or GluA1/A2 LBD layer, and in this respect is more akin to the GluN1b/N2B receptor.

The GluK2/K5em-CNQX density map allowed clear assignment of the three membrane-spanning helices (M1, M3, M4) on all four receptor subunits (*Figure 1A*, *Figure 1—figure supplement 7*). The small re-entrant helices (M2) were not resolved, likely because of conformational mobility. The TMD structure (*Figure 3A*) has a 'trapezoidal' shape and a square base (*Figure 3B*) which is seen in other members of the iGluR family (*Herguedas et al., 2019*; *Karakas and Furukawa, 2014*; *Kumari et al., 2019*; *Lee et al., 2014*; *Lü et al., 2017*; *Meyerson et al., 2016*; *Sobolevsky et al., 2009*). The antagonist-bound structure is expected to feature a closed ion channel; so to understand how GluK2/K5em restricts cation flow, we visualized the pore profile using HOLE (*Smart et al., 1996*; *Figure 3C–F*). This analysis revealed three constrictions on the M3 helices at T652/T636, A656/A640, and T660/T644 (GluK2em/GluK5em subunits). The positions of these constrictions match those seen for the GluK2 homomer (*Meyerson et al., 2016*) which reflects the strong sequence conservation for M3 helices within the KAR family. The availability of closed-channel structures for other iGluRs enabled a cross-family comparison of pore structure. We compared the pore profiles of GluK2/K5em-CNQX with closed-channel structures of a KAR homomer (GluK2, PDB: 5KUF), a di-heteromeric AMPAR (GluA1/A2, PDB: 6QKC), and a di-heteromeric NMDAR (GluN1b/N2B, PDB: 6WHU) (*Figure 3F*). These measurements highlight a similarity in pore structure for GluK2 and

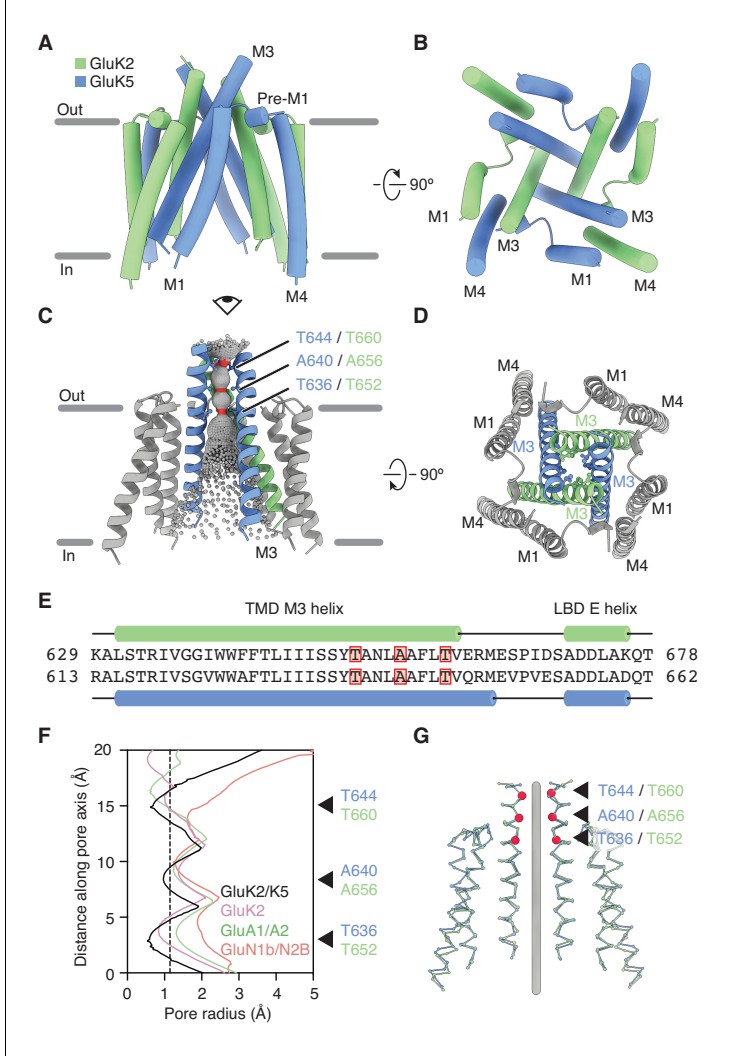

**Figure 3.** GluK2/K5 channel structure and symmetry. (**A and B**) GluK2/K5em transmembrane domain (TMD) as seen parallel to the membrane (**A**) and perpendicular to the membrane from the intracellular side (**B**). Eye icon in (**A**) gives the perspective presented in (**B**). GluK2em and GluK5em subunits rendered in green and blue, respectively. (**C**) Visualization of the pore volume. Regions with pore radius less than 1.15 Å are red, water-accessible sections with radius larger than 1.15 Å are gray. One receptor subunit is hidden to give a view of the pore. (**D**) View of the channel from the extracellular side with M3 helix bundle (ribbons) and residues at pore constriction sites (ball-and-stick atoms). (**E**) Sequence alignment between GluK2 and GluK5 for the M3 helix of the TMD and the E helix of the ligand binding domain (LBD). Residues at pore constrictions are highlighted in red. (**F**) Comparison of pore profiles for GluK2/K5em (black), GluK2 (magenta), GluA1/A2 (green), and GluN1/N2B (red). Arrows indicate constrictions along GluK2/K5em permeation pathway. Plot shows pore radius as a function of channel position. Vertical dashed line marks 1.15 Å pore radius. Analysis done using PDB files 5KUF (GluK2), 6QKC (GluA1/A2), and 6WHU (GluN1b/N2B). (**G**) The two GluK2em subunits are aligned as a rigid body with the two GluK5em subunits. Gray line marks the central axis of the channel. Red markers and arrows designate pore constrictions seen in (**C,F**).

GluK2/K5em and also show the KAR heteromer pore is more similar to the AMPAR than the NMDAR. Finally, given the mixed composition of the GluK2/K5em TMD, we asked how closely it adopts fourfold symmetry, given its mixed subunit composition. For this analysis we considered the correspondence between the pair of GluK2em and the pair of GluK5em subunits in the GluK2/K5em TMD (*Figure 3G*) and measured an RMSD of 1.5 Å. This near identity suggests that despite containing two different subunit types, the channel maintains near fourfold rotational symmetry.

## Desensitized GluK2/K5 is characterized by major GluK2 rearrangements

We next pursued a desensitized structure of GluK2/K5$_{em}$ to understand how the receptor may reorganize to close the channel in response to the sustained presence of L-Glu. Purified receptor was incubated with 1 mM L-Glu, imaged by cryo-EM, and the structure was solved by single particle analysis with no symmetry applied (*Figure 4—figure supplements 1* and *2*, and *Table 2*). The data was processed using a similar approach to that used for the antagonist-bound dataset, and we first resolved the full-length L-Glu-bound receptor to 5.8 Å, then independently resolved the ATD layer (3.8 Å) and LBD-TMD assembly (4.3 Å) (*Figure 4—figure supplement 1*). The resolution in the LBDs was not sufficient to resolve ligands, so we compared them to crystal structures of the GluK2 LBD bound by either L-Glu (PDB: 1S50) or a competitive antagonist (PDB: 5CMK) (*Figure 1—figure supplement 5B*). The structural alignments and the measured RMSD values showed unambiguously that the LBDs are similar to the L-Glu-bound LBD crystal structure.

The most striking feature of the GluK2/K5$_{em}$-L-Glu structure is the LBD layer, which shows rupture of both LBD dimer pairs and exhibits an apparent fourfold symmetric subunit arrangement (*Figure 4C,D*). Although the LBD layer gives the impression of fourfold symmetry, close inspection shows that the G helices of all four LBDs come together in a staggered arrangement to render the layer twofold symmetric (*Figure 4E*). Critically, this arrangement is reminiscent of the 'desensitization ring' observed in the desensitized GluK2 homomer structure (*Meyerson et al., 2016*). We compared the structure of desensitized GluK2/K5$_{em}$ to the structure of desensitized GluK2 (PDB: 5KUF) to establish their congruence (*Figure 4G–I*). The structures show good global agreement in the LBD layers, although the in-plane rotation for GluK2$_{em}$ (B/D) and GluK5$_{em}$ (A/C) LBDs differ with the rotations for their GluK2 homomer counterparts by ~10° (B/D) and 7° (A/C). Furthermore, close inspection shows GluK5$_{em}$ G helices reside ~3 Å closer to each other across the twofold symmetry plane of the LBD layer, compared to analogous GluK2 subunits in the homomer (*Figure 4E,H*). This subtle difference has a geometric consequence of producing a more compact desensitization ring in the heteromer (*Figure 4E,H*, insets).

We next wanted to understand the conformational differences between LBD layers in the GluK2/K5$_{em}$-L-Glu structure (*Figure 4E*) and in the GluK2/K5$_{em}$-CNQX structure (*Figure 1E*). The analysis showed that relative to the CNQX-bound structure, the L-Glu-bound GluK5$_{em}$ LBDs (A/C positions) are turned in plane by ~18°, while the GluK2$_{em}$ LBDs (B/D positions) are rotated by ~110° (*Figure 4A–C*). Meanwhile, the GluK5$_{em}$ LBDs pitch back by ~10° more than GluK2 partner subunits (*Figure 4F*, inset), while both GluK2$_{em}$ and GluK5$_{em}$ show the same change in cleft closure (~20°) between their open-cleft (CNQX) and closed-cleft (L-Glu) states. Critically, this latter measurement confirms that while the resolution of the desensitized structure is not sufficient to resolve ligand density, all four LBDs have closed clefts consistent with agonist binding (*Møllerud et al., 2017*; *Figure 1—figure supplement 5C*). Overall this analysis shows that CNQX-bound and L-Glu-bound structures have radically different LBD configurations. While the CNQX-bound structure shows intact LBD dimers, the L-Glu bound structure shows ruptured and reorganized LBDs where all four LBD binding clefts are directed toward the receptor central axis. This is well illustrated by a structural morph between the two structures (*Videos 1* and *2*). The LBD organization in the desensitized state is entirely consistent with the model where L-Glu binding causes LBDs to open the channel, and the LBD dimer pairs then rupture to allow the channel to close (*Armstrong et al., 2006*; *Dürr et al., 2014*; *Meyerson et al., 2014a*; *Twomey et al., 2017*).

## The M3-S2 linkers facilitate two different closed-channel receptor states

KARs function with a resting state with LBDs poised to bind L-Glu and activate the channel, and a desensitized state with LBDs bound to L-Glu after activation concludes. Critically, both states have a closed ion channel. Our structures of GluK2/K5$_{em}$ show that apo resting and antagonist-bound states have intact LBD dimers arranged with twofold local symmetry (*Figure 1E*, *Figure 1—figure supplement 4*), while the desensitized state has ruptured LBD dimers arranged with pseudo-fourfold local symmetry but true twofold local symmetry (*Figure 4D*). This raises the question of how the receptor can accommodate such radically different LBD conformations in two closed-channel structures.

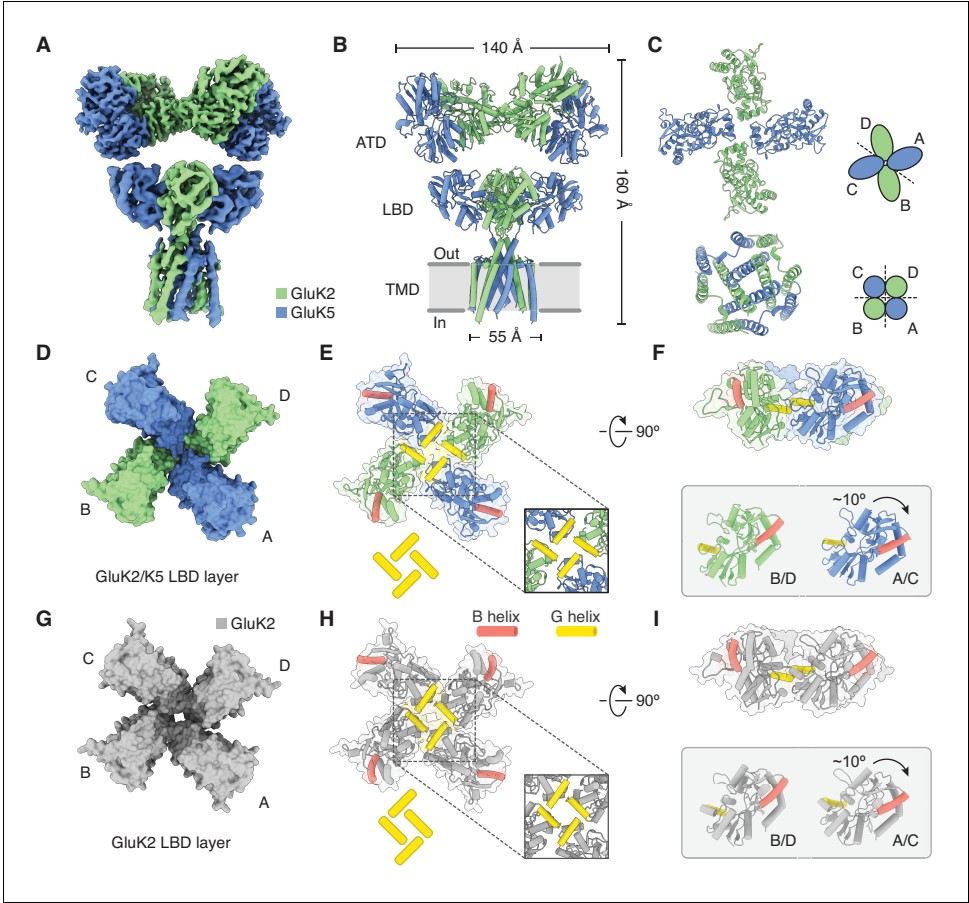

**Figure 4.** Structure of GluK2/K5 in a desensitized state. (**A**) Cryo-electron microscopy (cryo-EM) structure of the GluK2/K5em heteromer in a L-glutamate (L-Glu)-bound state with GluK2em and GluK5em subunits rendered in green and blue, respectively. The panel shows the map for the amino terminal domain (ATD) layer and the map for the ligand binding domain (LBD)-transmembrane domain (TMD) assembly which were reconstructed independently, as described in the text. (**B**) Molecular model for the receptor is colored as in (**A**) and shown parallel to the membrane. (**C**) The LBD (top) and TMD (bottom) layers as viewed from the extracellular space. The local symmetries of the LBD (twofold) and TMD (fourfold) are illustrated. (**D and E**) Top view of LBD layer for the GluK2/K5em heteromer. The model is shown without (**D**) and with (**E**) secondary structure. Helices B (red) and G (yellow) are colored as visual guides for subunit orientation. Inset highlights the desensitization ring formed by G helices. (**F**) LBDs are shown colored as in (**E**) but viewed parallel to the membrane. Individual B/D and A/C LBDs are aligned and presented to highlight different out of plane tilts, and the more elevated position of the G helices in A/C subunits. (**G–I**) Visualization of the LBD layer from the GluK2 homomer (PDB: 5KUF). GluK2 subunits shown in gray, and panel coloring and arrangement is otherwise the same as in (**D–F**).

The online version of this article includes the following figure supplement(s) for figure 4:

**Figure supplement 1.** Cryo-electron microscopy (cryo-EM) image processing workflow for GluK2/K5em with L-glutamate (L-Glu).

**Figure supplement 2.** Resolution determination for GluK2/K5em structures with L-glutamate (L-Glu).

To address this question we compared antagonist-bound and desensitized structures, and on both structures analyzed the region spanning the M3 helices (the major pore-forming helices in the TMD), the M3-S2 linkers (which join the M3 helices and LBD), and the E helices (the lower lobe of the LBD which couples to M3) (*Figure 5—figure supplement 1*). We first considered the overall correspondence between GluK2/K5em-CNQX and GluK2/K5em-L-Glu pores. The M3 helices (L631/L615 to M664/M648 for GluK2em/GluK5em) in both structures are similar with an RMSD of 2.8 Å. The similarity in M3 helices indicates that the LBD-TMD linkers alone accommodate the two distinct LBD layer configurations. To understand how this is accomplished, we isolated the lateral (*Figure 5D,E*) and vertical (*Figure 5F,G*) components of M3-S2 linker positions and considered differences in

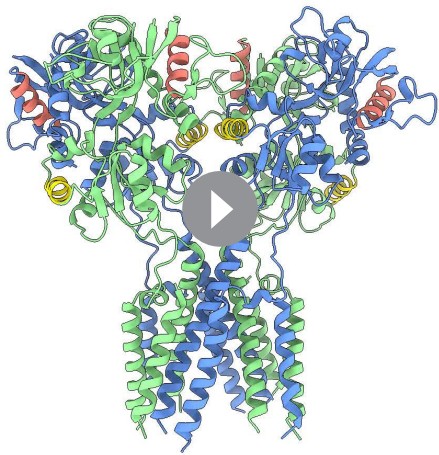

**Video 1.** Morph for GluK2/K5 ligand binding domain (LBD)-transmembrane domain (TMD) assembly viewed from the side. The morph presents the conformational differences between antagonist-bound and desensitized states of GluK2/K5$_{em}$. The LBD and TMD layers are presented without the amino terminal domain (ATD) layer and as viewed parallel to the membrane. The GluK2$_{em}$ and GluK5$_{em}$ subunits are shown in green and blue, respectively. B helices are shown in red, and G helices in yellow.

https://elifesciences.org/articles/66097#video1

alpha-carbon position. For analysis of lateral linker positions, we noted that the four linkers arrange with clear twofold symmetry in the antagonist-bound state but adopt near fourfold symmetry in the desensitized state (*Figure 5D,F*, and *Videos 3* and *4*). Accordingly we measured the extent to which each chain in the desensitized structure deviates from its counterpart in the antagonist-bound structure (*Figure 5E*). Through this calculation we identified residue E662 on the M3 helices of GluK2$_{em}$ as the key site where the antagonist-bound and agonist-bound chains diverge (*Figure 5E*, arrow).

We analyzed the vertical component of the M3-S2 linker conformations by measuring the vertical position of each alpha-carbon ranging from the intracellular end of the M3 helix up through helix E at the base of the LBD (*Figure 5F*). These values were plotted as a function of residue number for all four subunits in both CNQX-bound and L-Glu-bound structures (*Figure 5G*). This vertical analysis revealed that GluK5$_{em}$ (A/C) linker elevation does not differ significantly between antagonist-bound and desensitized conformations. This is consistent with the overall observa-

tion that these subunits do not show major conformational differences between the two states. Meanwhile, for GluK2$_{em}$ (B/D) subunits, the M3-S2 linkers extend ~12 Å further from the membrane in the desensitized state. This latter measurement suggests that to facilitate channel closure during desensitization, the GluK2 subunits may move away from the membrane in addition to undergoing large in-plane rotations. This analysis yields the conclusion that the M3-S2 linkers on the GluK2 subunit are the core structural element that allows GluK2/K5 to adopt two significantly different LBD arrangements in the antagonist-bound and desensitized states.

## Discussion

Defining how the subunits of GluK2/K5 organize and respond to neurotransmitter has been a major research goal since the discovery of the receptor 30 years ago (*Herb et al., 1992*). In our study we show that the heteromer assembles with two copies of each subunit, with GluK2 and GluK5 LBDs arranged in alternating fashion around the receptor central axis and with GluK5 subunits proximal to the pore and GluK2 subunits

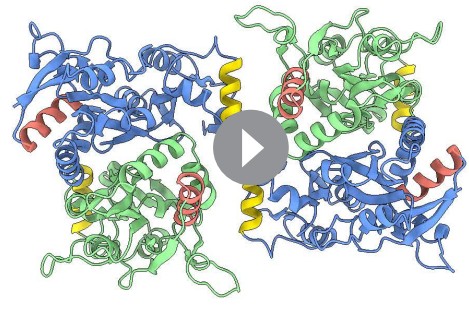

**Video 2.** Morph for GluK2/K5$_{em}$ ligand binding domain (LBD)-transmembrane domain (TMD) assembly viewed from the top. The morph presents the conformational differences between antagonist-bound and desensitized states of GluK2/K5$_{em}$. The LBD layer is presented without the amino terminal domain (ATD) or TMD layers, and as viewed perpendicular to the membrane from the extracellular space. The GluK2$_{em}$ and GluK5$_{em}$ subunits are shown in green and blue, respectively. B helices are shown in red, and G helices in yellow.

https://elifesciences.org/articles/66097#video2

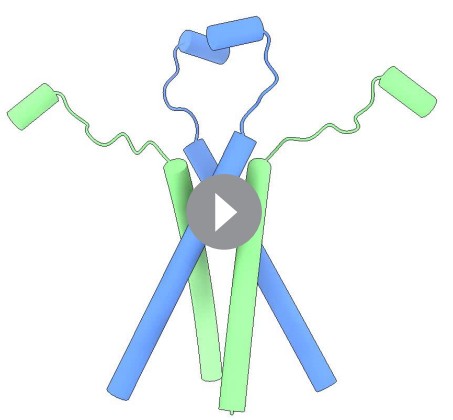

**Video 3.** Morph of M3 and E helices viewed from side. The movie presents conformational differences in M3 transmembrane helices and E helices of the ligand binding domains (LBDs) between antagonist-bound and desensitized states. The perspective is parallel to the membrane. The GluK2$_{em}$ and GluK5$_{em}$ subunits are shown in green and blue, respectively.

https://elifesciences.org/articles/66097#video3

during its slow deactivation. Another avenue to such a structure would be to visualize GluK2/K5 in the presence of the ligand AMPA, and exploit the fact that GluK5, and not GluK2, binds AMPA to produce stable currents (*Herb et al., 1992*).

This study shows that the LBD layer of desensitized GluK2/K5 is organized with pseudo-fourfold symmetry and with the four LBDs positioned such that their G helices are in a staggered arrangement near the central axis of the receptor (*Figure 4E*). This LBD configuration is similar to that observed in desensitized GluK2 (*Figure 4H*; *Meyerson et al., 2016*) and GluK3 (*Kumari et al., 2019*), which suggests the arrangement is a hallmark of desensitization in the KAR family. The mechanistic role of this LBD arrangement cannot yet be fully understood because no active state KAR structure is available. However, it is clear that desensitization must involve reorganizing the LBD dimer interface to allow LBDs to remain bound to L-Glu while also closing the channel (*Figure 1—figure supplement 2*; *Armstrong et al., 2006*; *Twomey and Sobolevsky, 2018*; *Zhu and Gouaux, 2017*). The LBD layer motif we observe would satisfy these requirements with its ruptured LBD dimers and LBDs arranged to accommodate a closed ion channel. Furthermore, the

distal to the pore (*Figure 1*). The structure shows that GluK2 and GluK5 LBDs assemble as heterodimers, validating what was hypothesized from previous studies of GluK2/K5 (*Kristensen et al., 2016*; *Kumar et al., 2011*; *Litwin et al., 2020*; *Paramo et al., 2017*; *Reiner et al., 2012*).

GluK5 is distinguished from GluK2 in that it has high L-Glu affinity and slow deactivation kinetics, which sustains non-desensitizing currents in GluK2/K5 heteromers (*Barberis et al., 2008*). Because of the high affinity of GluK5 for L-Glu, at low neurotransmitter concentration GluK2/K5 is activated by ligand binding to the GluK5 subunits without binding to GluK2 subunits (*Fisher and Mott, 2011*). Functional studies have found that desensitization is triggered when GluK2 subunits are also bound by neurotransmitter (*Fisher and Fisher, 2014*; *Reiner and Isacoff, 2014*). The desensitized GluK2/K5 structure helps rationalize this functional role for GluK2, suggesting that it is the GluK2 subunits which undergo large conformational changes to facilitate channel closure during desensitization (*Figure 4*). The fact that GluK5 can sustain non-desensitizing currents at low-micromolar L-Glu concentration (*Fisher and Mott, 2011*) presents the enticing potential to resolve an active state KAR structure, and also sheds light on how GluK2/K5 may sustain current

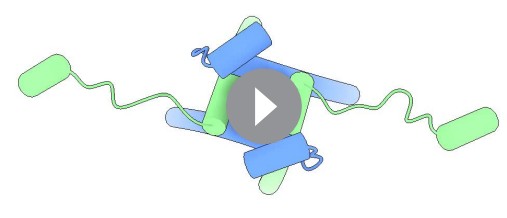

**Video 4.** Morph of M3 and E helices viewed from top. The movie presents conformational differences in M3 transmembrane helices and E helices of the ligand binding domains (LBDs) between antagonist-bound and desensitized states. The perspective is perpendicular to the membrane from the extracellular space. The GluK2$_{em}$ and GluK5$_{em}$ subunits are shown in green and blue, respectively.

https://elifesciences.org/articles/66097#video4

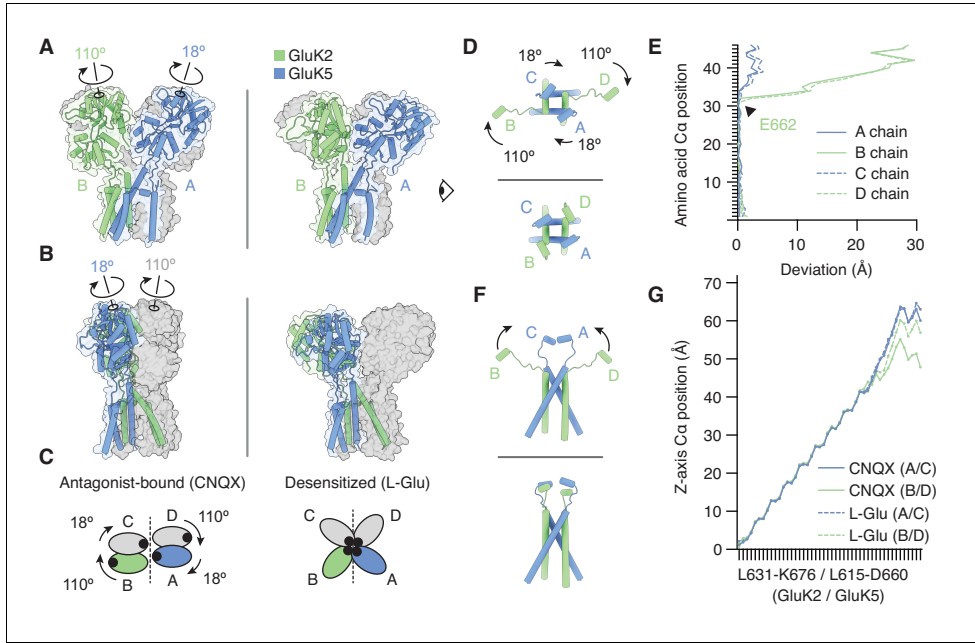

**Figure 5.** The M3-S2 linkers accommodate different ligand binding domain (LBD) arrangements. (**A**) GluK2/K5$_{em}$ LBD-transmembrane domain (TMD) assembly, without the amino terminal domain (ATD) layer. Antagonist-bound state (left) annotated with arrows to convey conformational differences with the desensitized state (right). (**B**) Conformational differences as shown from the perspective of the eye icon given in (**A**). (**C**) Cartoon depicting differences between the states as viewed extracellularly, down the receptor axis. Black dots symbolize LBD binding cleft locations. (**D and E**) Lateral motions along the range from M3 helices through E helices. Comparison between antagonist-bound (**D**, top) and desensitized (**D**, bottom). Panel (**E**) considers the antagonist-bound state as a reference and measures the extent to which each chain in the desensitized state deviates from that reference. M3 helices show low deviation in all four subunits, consistent with minimal conformational difference. The M3-S2 linkers and E helix show modest deviation in GluK5$_{em}$ (A/C) and large deviations in GluK2$_{em}$ (B/D). Ranges are from L631-K676 (GluK2$_{em}$) and L615-D660 (GluK5$_{em}$). (**F and G**) Vertical differences along the M3 helix through E helix range. Comparison between antagonist-bound (**F**, top) and desensitized (**F**, bottom) indicates that lateral differences in GluK2$_{em}$ (B/D) subunits position are paired with a vertical rise (~6 Å). Panel (**G**) measures the vertical rise in each subunit as a function of amino acid residue, showing that for the antagonist-bound state the GluK2$_{em}$ (B/D) E helices reside at lower positions than in GluK5$_{em}$ subunits (A/C). This difference is essentially not present in the desensitized state, where all subunit chains rise in similar ways to reach approximately equal elevations above the membrane.

The online version of this article includes the following figure supplement(s) for figure 5:

**Figure supplement 1.** Cryo-electron microscopy (cryo-EM) density and model for M3 helices, M3-S2 linkers, and E helices.

E662 position at the top of the M3 helices on GluK2 may serve as a fixed point around which GluK2 LBDs and M3-S2 linkers rotate during desensitization (*Figure 5E*).

Previous electrophysiology of KARs and X-ray crystallography of isolated LBDs have demonstrated that KAR homomers are regulated by extracellular ions that bind to the LBDs (*Dawe et al., 2013*; *Plested and Mayer, 2007*; *Plested et al., 2008*). However, a recent molecular dynamics study suggests that the canonical view of ion-dependent gating in homomeric receptors (*Dawe et al., 2013*) is different in GluK2/K5 heteromers (*Paramo et al., 2017*). Specifically, while sodium stabilizes GluK2 homomers, with a weaker allosteric effect by lithium ions, the situation is reversed in GluK2/K5 heteromers where lithium potentiates the agonist response and prolongs the rate of desensitization. Although we could not address questions of ion binding because of the resolution in our cryo-EM maps, the experimental foundation established in this study along with recent advances in atomic resolution cryo-EM (*Nakane et al., 2020*; *Yip et al., 2020*) pave the way for the future study of how heteromerization and subunit placement affects anion and cation regulation of GluK2/K5 heteromers. Indeed, further work which combines high-resolution cryo-EM with electrophysiology

will make it possible to better understand the full extent to which the ion regulatory mechanism differs between homomeric and heteromeric channels.

It is notable that after refining the full-length structures of CNQX-bound and L-Glu-bound GluK2/K5$_{em}$, we found that the resolutions of the ATD layer and the LBD-TMD assembly could be improved with local refinement (*Figure 1—figure supplement 7*, *Figure 4—figure supplement 2*). This suggests some degree of relative mobility between these regions permitted by the ATD-LBD linkers. The ATDs in both of these structures also gave the highest resolutions in the study so we searched for local conformational differences that might point to novel functional attributes of the ATDs. However, close comparison showed no significant differences between the two ATD layers which is consistent with the idea that KAR ATDs are not involved in channel gating. Rather, they are thought to play a key role in subunit oligomerization during receptor biogenesis (*Kumar et al., 2011*; *Zhao et al., 2017*).

A key question during this study was whether GluK5 is always found at the A/C positions, or might other assemblies exist. Indeed, we performed extensive image classification in the hopes of discovering receptors that did not obey the proposed assembly. Despite this effort we did not find evidence that such particle subsets exist in the datasets. Importantly, given the 10-fold excess of GluK5$_{em}$ versus GluK2$_{em}$ virus added during protein expression, we would expect GluK5 to adopt alternative positions (i.e. B or D) if this were possible. That this does not happen led us to conclude that GluK5 can only occupy the A/C positions.

We note that this study used GluK2 and GluK5 subunits that were modified to facilitate protein expression and purification (*Figure 1—figure supplement 1*, *Figure 1—figure supplement 2A*), and while these subunits display similar functional properties to wild-type subunits (*Figure 1—figure supplement 2* and *Table 1*), it will be valuable to test the observed subunit arrangement using an orthogonal approach. To this end the subunit arrangement seen in the structures will be supported by future experiments using a technique such as crosslinking to reinforce the conclusion that it represents the predominant physiological form of GluK2/K5.

This study marks a milestone in iGluR biology by providing the first structures of the presumed major neuronal type of KAR to accompany structures of the dominant forms of NMDARs and AMPARs. The structures answer long-standing questions about domain arrangements and orientations, and they lay a foundation for high-resolution studies of GluK2/K5 to fully illuminate its gating mechanism and coordination of small-molecule agonists, antagonists, and modulators (*Jane et al., 2009*; *Larsen et al., 2017*). Together the availability of molecular models for all iGluR family members now makes possible comparative structure-guided investigation across the iGluR family and a framework for precision targeting of drugs to specific receptor subtypes.

# Materials and methods

## Key resources table

| Reagent type (species) or resource | Designation | Source or reference | Identifiers | Additional information |
|---|---|---|---|---|
| Gene (*Rattus norvegicus*) | GRIK2_RAT | Provided by Dr. Janet Fisher (University of South Carolina) | P42260 | |
| Gene (*Rattus norvegicus*) | GRIK5_RAT | Provided by Dr. Janet Fisher (University of South Carolina) | Q63273 | |
| Cell line (*Homo sapiens*) | HEK293S GnTI⁻ | ATCC | ATCC, Cat. No. CRL-3022 | |
| Cell line (*Homo sapiens*) | HEK293T/17 | ATCC | ATCC, Cat. No. CRL-11268 | |
| Cell line (insect) | Sf9 | Expression Systems, courtesy of Dr. Xin-Yun Huang (Weill Cornell Medical College) | Expression Systems, Cat. No. 94–001S | |

*Continued on next page*

*Continued*

| Reagent type (species) or resource | Designation | Source or reference | Identifiers | Additional information |
|---|---|---|---|---|
| Recombinant DNA reagent | pEZTBM vector | doi: 10.1016/j.str.2016.03.004 | | https://www.addgene.org/74099/ |
| Chemical compound, drug | 6-Cyano-7-nitroquinoxaline-2,3-dione (CNQX) | Tocris | Tocris, Cat. No. 1045 | |
| Chemical compound, drug | L-Glutamate | Sigma | Sigma, Cat. No. G1251 | |
| Software, algorithm | pClamp 9, Clampfit 10.5 | Molecular Devices, LLC | RRID:SCR_011323 | http://www.moleculardevices.com/products/software/pclamp.html |
| Software, algorithm | OriginPro 2020 | OriginLab | RRID:SCR_014212 | https://www.originlab.com/2020 |
| Software, algorithm | Relion 3.1 | doi:10.7554/eLife.42166 | RRID:SCR_016274 | https://www3.mrc-lmb.cam.ac.uk/relion/index.php/Main_Page |
| Software, algorithm | cryoSPARC 2 | doi:10.1038/nmeth.4169. | RRID:SCR_016501 | https://cryosparc.com/ |
| Software, algorithm | CTFFIND 4.1 | doi:10.1016/j.jsb.2015.08.008 | RRID:SCR_016732 | http://grigorieflab.janelia.org/ctffind4 |
| Software, algorithm | UCSF Chimera | doi:10.1002/jcc.20084 | RRID:SCR_004097 | http://plato.cgl.ucsf.edu/chimera/ |
| Software, algorithm | UCSF ChimeraX | doi:10.1002/pro.3943 | RRID:SCR_015872 | https://www.cgl.ucsf.edu/chimerax/ |
| Software, algorithm | Bsoft | doi:10.1006/jsbi.2001.4339 | RRID:SCR_016503 | https://lsbr.niams.nih.gov/bsoft/ |
| Software, algorithm | HOLE | doi:10.1016/s0263-7855(97)00009-x | | http://www.holeprogram.org |
| Software, algorithm | COOT 0.9 | doi:10.1107/S0907444910007493 | RRID:SCR_014222 | http://www2.mrc-lmb.cam.ac.uk/personal/pemsley/coot/ |
| Software, algorithm | Phenix 1.14 | doi:10.1107/S2059798319011471 | RRID:SCR_014224 | https://www.phenix-online.org/ |
| Software, algorithm | GraphPad Prism | GraphPad | RRID:SCR_002798 | http://www.graphpad.com/ |
| Software, algorithm | MolProbity | doi:10.1107/S0907444909042073 | RRID:SCR_014226 | http://molprobity.biochem.duke.edu |

## Construct design

Heteromer expression constructs were designed starting with genes for full-length rat GluK2 and GluK5. The genes were cloned into the pEZT-BM vector (*Morales-Perez et al., 2016*) and fused in frame via a thrombin recognition site to a Twin-Strep affinity tag (GluK2), and EGFP and a 1D4 tag (GluK5). The GluK2 gene underwent RNA editing at position 567 (I to V), and mutation of C576V and C595S to promote subunit expression (*Schauder et al., 2013*). The GluK5 gene was mutated at four cysteine positions (C559V, C578S, C619I, and C813A), truncated at position 827 after the M4 helix, and the GluA2 'tail' (YKSRAEAKRMK) (*Lee et al., 2014*; *Song et al., 2018*) was added at the subunit C-terminus to improve heteromer monodispersity. The constructs are referred to as GluK2$_{em}$ and GluK5$_{em}$ and their heteromeric complex as GluK2/K5$_{em}$.

## Electrophysiology

HEK293T cells (ATCC CRL-11268) were plated at low density ($1.6 \times 10^4$ cells/mL) on poly-D-lysine-coated 35 mm dishes and transiently transfected 24 hr post-plating using the calcium phosphate precipitation method. For functional tests of GluK2$_{em}$ and GluK5$_{em}$, the constructs were expressed from the pEZT-BM vector. Other experiments used GluK2 and GluK5 cDNAs expressed from the pRK5 plasmid and each construct contained a downstream IRES sequence encoding mCherry or EGFP, respectively. A GluK2:GluK5 ratio of 1:10 was used for all co-transfections. After 12–16 hr, cells were washed with divalent PBS and maintained in fresh medium (MEM containing GlutaMAX supplemented with 10% fetal bovine serum).

Recordings were performed 36–48 hr post-transfection on outside-out patches excised from transfected cells. L-Glu (1 mM) was applied using a piezo-stack-driven perfusion system (Physik Instrumente). Solution exchange (<400 µs) was determined in a separate experiment by measuring the liquid junction current. The holding potential during recordings was −60 mV. External solution contained (in mM): 150 NaCl, 5 HEPES, 0.1 CaCl$_2$ and 0.1 MgCl$_2$, and 2% phenol red at pH 7.3–7.4. Internal solution contained (in mM): 115 NaCl, 10 NaF, 5 HEPES, 5 Na$_4$BAPTA, 0.5 CaCl$_2$, 1 MgCl$_2$,

and 10 Na$_2$ATP at pH 7.3–7.4. The osmotic pressure of all solutions was adjusted to 295–300 mOsm with sucrose. Recording pipettes were composed of borosilicate glass (3–6 MΩ, King Precision Glass, Inc) coated with dental wax. All recordings were performed at room temperature using an Axopatch 200B amplifier (Molecular Devices, LLC). Current records were filtered at 5 kHz and sampled at 25 kHz. Series resistance (3–12 MΩ) was compensated for by 95%. Data were acquired using pClamp nine software (Molecular Devices, LLC) and analyzed using Clampfit 10.5 (Molecular Devices, LLC). Data were visualized using OriginPro 2020 (OriginLab). A minimum of three individual patch recordings per condition were collected. Individual patch recordings were collected from multiple transfections.

## Protein expression and purification

The GluK2/K5$_{em}$ heteromer was expressed using the BacMam method (*Goehring et al., 2014*). Protein expression constructs were transformed into DH10Bac cells to produce bacmids. Bacmids for both constructs were transfected into Sf9 cells grown in ESF 921 media (Expression Systems). P1 and P2 virus production was monitored using GFP fluorescence from the pEZT-BM vector until virus harvesting. HEK293S GnTI$^-$ cells (ATCC CRL-3022) were grown (3.2 or 6.4 L) at 37°C and 8% CO$_2$ to a density of $3.5 \times 10^6$ cells/mL in FreeStyle suspension media (Gibco) supplemented with 2% fetal bovine serum (Gibco) and Anti-Anti (Gibco). P2 viruses for GluK2 and GluK5 were added to cells at a ratio of 1:10 (GluK2$_{em}$:GluK5$_{em}$) for a combined 10% (v/v) of the cell suspension. The suspension was incubated at 37°C for 24 hr, then sodium butyrate (Sigma) was added to a final concentration of 10 mM and flasks were shifted to 30°C and 8% CO$_2$. Cells were collected 84 hr after transduction by low-speed centrifugation, flash-frozen in liquid nitrogen, and stored at −80°C. Cell pellets were resuspended in ice-cold resuspension buffer containing 20 mM Tris (pH 8.0), 300 mM NaCl, 0.8 μM aprotinin, 2 μg/mL leupeptin, 2 μM pepstatin, 0.5 mM EDTA, and 1 mM PMSF (1 mL buffer per 1 g of cell pellet) and manually pipetted until no clumps remained. An equal volume of solubilization buffer containing 20 mM Tris (pH 8.0), 300 mM NaCl, supplemented with 100 mM dodecyl-β-D-maltoside (DDM, Anatrace), 0.5% cholesteryl hemisuccinate (CHS, Anatrace), 0.8 μM aprotinin, 2 μg/mL leupeptin, 2 μM pepstatin, 0.5 mM EDTA, and 1 mM PMSF was added to the mixture, and the sample was nutated for 90 min at 4°C. The mixture was spun by low-speed centrifugation at 20,000×*g* for 20 min, followed by ultracentrifugation at 125,000×*g* for 90 min. The supernatant was filtered through a 0.45 μm filter to remove debris, bound to a Strep-Tactin column (GE) equilibrated with running buffer (20 mM Tris, pH 8.0, 300 mM NaCl, 1 mM DDM, 0.005% CHS), washed with 10 column volumes of running buffer, and eluted in running buffer supplemented with 20 mM desthiobiotin (IBA). The elution fraction from Twin-Strep affinity purification was then mixed with 1D4 affinity resin (Cube Biotech) equilibrated in running buffer and bound in batch mode for 1 hr at 4°C with gentle mixing. The resin was washed with 100 column volumes of running buffer and eluted using running buffer supplemented with 0.2 mM 1D4 peptide (Cube Biotech). The receptor was concentrated and loaded onto a Superose 6 Increase 10/300 GL column equilibrated with gel filtration buffer containing 20 mM Tris (pH 8.0), 300 mM NaCl, 0.35 mM DDM, 0.0017% CHS. Elution fractions were collected, analyzed by SDS-PAGE, and peak fractions were concentrated and used for cryo-EM experiments. Protein biochemistry experiments were done with several batches of cells and different virus infections during optimization to ensure reproducibility. The full-scale purification of GluK2/K5$_{em}$ was performed successfully three times.

## Cell lines

In this study we used authentic and mycoplasma-free cells that were provided by Expression Systems and ATCC, and we did not do additional authentication or mycoplasma testing. At the time of purchase, the cells were frozen into multiple stocks. After approximately 30 passages (Sf9) or 20 passages (HEK), a new vial of cells was thawed for use and the previous batch discarded.

## Cryo-EM sample preparation and data acquisition

Samples were prepared using GluK2/K5$_{em}$ receptor (3.7 mg/mL) without ligand (apo), or receptor incubated with 1 mM CNQX (Tocris) or 1 mM L-Glu (Sigma). UltrAuFoil 1.2/1.3 300 mesh grids (Quantifoil) were plasma-treated and rendered hydrophilic by reaction with PEG-thiol (*Meyerson et al., 2014b*). Vitrified samples were prepared by adding a 2.5 μL droplet of sample

solution to a grid, then blotting (2 s blot time, 0 or −1 blot force) and plunge-freezing in liquid ethane using a Vitrobot Mk IV (Thermo Fisher).

Single particle images of GluK2/K5$_{em}$-apo were collected with Leginon (*Suloway et al., 2005*) using a Titan Krios electron microscope (Thermo Fisher) operated at 300 kV and a nominal magnification of 81,000 × and equipped with a GIF (Gatan) and K3 camera (Gatan) set in counted mode (1.083 Å pixel size). Exposures were made with an average defocus value of ~1.6 µm, dose fractionation into 40 frames, a total exposure time of 2 s, and total dose of 51.23 e⁻/Å². A total of 970 movies were recorded.

Single particle images of GluK2/K5$_{em}$-CNQX and GluK2/K5$_{em}$-L-Glu were collected with Leginon (*Suloway et al., 2005*) using an Arctica electron microscope (Thermo Fisher) operated at 200 kV and a nominal magnification of 36,000× and equipped with a K3 camera (Gatan) set in super-resolution mode (0.5480 Å pixel size). Exposures were done with nominal defocus values between 0.4 and 4.8 µm, dose fractionation into 40 frames, a total exposure time of 2.8 s, and total dose of 50–53 e⁻/Å². A total of 9076 (GluK2/K5$_{em}$-CNQX) and 5275 (GluK2/K5$_{em}$-L-Glu) movies were recorded.

## Image processing and structural analysis

Movie stacks were corrected for beam-induced motion (with twofold binning for super-resolution K3 data) in Relion 3.1 (*Zivanov et al., 2018*) yielding dose-weighted images with an image pixel size of 1.083 Å for the apo state dataset, and 1.096 Å for CNQX-bound and L-Glu-bound datasets. These images were used for contrast transfer function (CTF) estimation with CTFFIND4.1 (*Rohou and Grigorieff, 2015*). Reference-free particle auto-picking was done using Laplacian-of-Gaussian tool in Relion. The auto-picked particles were extracted with a box size of 416 pixels, binned to a box size of 64 or 128 pixels, and imported into cryoSPARC (*Punjani et al., 2017*). One round of 2D classification was performed for CNQX-bound and L-Glu-bound datasets but was not used for the apo dataset. Ab initio 3D reconstruction and several rounds of heterogeneous refinement with C1-symmetry were used for all three datasets. This process removed false positives from auto-picking and isolated interpretable particle sets (*Figure 1—figure supplements 4* and *6*, and *Figure 4—figure supplement 1*).

The resulting particles were re-extracted with a box size of 320 or 416 without binning and imported into cryoSPARC. Particles extracted with a box size of 320 pixels were first re-centered. Ab initio 3D reconstruction and several rounds of heterogeneous refinements were again performed with C1-symmetry to obtain the final set of particles. The final 3D reconstructions were obtained by non-uniform (GluK2/K5$_{em}$-apo) or homogeneous refinement (GluK2/K5$_{em}$-CNQX and GluK2/K5$_{em}$-L-Glu) in cryoSPARC. The global resolutions were calculated using the FSC as 7.5 Å (apo), 5.3 Å (CNQX), and 5.8 Å (L-Glu) (*Figure 1—figure supplements 4* and *7*, and *Figure 4—figure supplement 2*, *Table 2*). Local resolution was calculated using ResMap (*Kucukelbir et al., 2014*) and visualized in UCSF ChimeraX (*Goddard et al., 2018*).

The GluK2/K5$_{em}$-CNQX ATD layer and LBD-TMD assembly were independently refined starting with particles from homogeneous refinement of the full-length receptor. For the ATD layer, The LBD-TMD signal was subtracted from the particles in Relion, then the particles were processed with ab initio reconstruction, heterogeneous refinement, non-uniform refinement, and local refinement in cryoSPARC to a resolution of 3.6 Å (*Figure 1—figure supplements 6* and *7*, and *Table 2*). For the LBD-TMD assembly, the particles were subjected to 2D classification, ab initio reconstruction, and several rounds of heterogeneous refinements in cryoSPARC to further isolate a particle subset. A round of non-uniform refinement was performed followed by particle subtraction to remove the ATD layer. The signal-subtracted particles were then processed with 2D classification, ab initio reconstruction, and several rounds of heterogeneous refinements to isolate a well-resolved LBD-TMD class. Non-uniform refinement (C2 symmetry) was used in a final step to refine the assembly to 4.2 Å (*Figure 1—figure supplements 6* and *7*, *Table 2*). C1 symmetry was used throughout unless otherwise noted.

The GluK2/K5$_{em}$-L-Glu ATD layer and LBD-TMD assembly were separately refined starting with the particles used for homogeneous refinement of the full-length receptor, and which had been further processed by a round of 2D classification and non-uniform refinement in cryoSPARC. To refine the ATD layer, particles were subjected to signal subtraction to remove the LBD-TMD region, then processed with ab initio reconstruction and several rounds of heterogeneous refinements to isolate a uniform population of ATD particles. Non-uniform refinement (C2 symmetry) followed by local

refinement were used to obtain an ATD structure at 3.8 Å resolution (*Figure 4—figure supplements 1* and *2*, *Table 2*). The LBD-TMD assembly was resolved by subtracting ATD signal from the particles, followed by 2D classification, ab initio reconstruction, and several rounds of heterogenous refinements to obtain a uniform particle subset, and then a final round of non-uniform refinement (C2 symmetry) gave a resolution of 4.3 Å (*Figure 4—figure supplements 1* and *2*, *Table 2*). C1 symmetry was used throughout unless otherwise noted.

Additional analysis was performed in an attempt to improve resolutions in the LBD and TMD regions of both GluK2/K5$_{em}$-CNQX and GluK2/K5$_{em}$-L-Glu, and we first attempted to use masked local refinement of these regions. This strategy did not improve the resolutions likely because of the small masses of the LBD and TMD layers, so we next applied conformational variability analysis in cryoSPARC. This showed no evidence for local conformational variability in either the LBD or TMD regions.

### Structural modeling

To model GluK2/K5$_{em}$-CNQX and GluK2/K5$_{em}$-L-Glu, the ATD and LBD-TMD maps were first aligned to their respective full-length maps using Chimera (*Pettersen et al., 2004*). This made sure that ATD and LBD-TMD maps were correctly positioned relative to each other. The ATD and LBD-TMD maps were then used for model building.

The ATD cryo-EM maps were resolved to 3.6 Å (GluK2/K5$_{em}$-L-CNQX) and 3.8 Å resolution (GluK2/K5$_{em}$-L-Glu) which permitted side chain modeling. The ATDs were modeled starting from the GluK2/K5 ATD crystal structure (PDB: 3QLU) which was docked into the maps using Chimera. The model was rebuilt using COOT (*Emsley et al., 2010*) and refined using Phenix (*Liebschner et al., 2019*).

The LBD-TMD assemblies were built starting from a homology model based on the GluK2 homomer structure (PDB: 5KUF) and generated by the SWISS-MODEL server (*Waterhouse et al., 2018*). The LBD and TMD homology models were docked into the LBD-TMD maps using Chimera, then rebuilt using COOT. Because the global resolutions of the LBD-TMD maps were 4.2 Å (CNQX) and 4.3 Å (L-Glu) (*Table 2*), the assemblies were modeled as poly-alanine chains. The exception to this was the pore-lining M3-helix bundle crossing in the TMD where side chains were visible in both CNQX-bound and L-Glu-bound maps and could be modeled for structural analysis (*Figure 5C–G*, *Figure 5—figure supplement 1*). Specifically, side chains were modeled at positions 620–646 (A/C, GluK5 subunits) and 636–661 (B/D, GluK2 subunits). The ATD-LBD linkers, S2-M4 linkers, and M2 helices were not resolved in either structure and were not modeled. In addition, the S1-M1 linkers in GluK2/K5$_{em}$-L-Glu were not resolved or modeled.

## Acknowledgements

We thank J Fisher (University of South Carolina) for providing genes for GluK2 and GluK5, and N Panitz (SUNY Binghamton) for assistance with image processing pipeline development. We thank members of our labs, O Clarke (Columbia University), and the A Accardi, O Boudker, and XY Huang labs for discussions. Electron microscopy was performed at the NYU Langone Health's Cryo–Electron Microscopy Laboratory with assistance from WJ Rice and B Wang, and at the Simons Electron Microscopy Center and National Resource for Automated Molecular Microscopy located at the New York Structural Biology Center and supported by grants from the Simons Foundation (SF349247), NYSTAR, and the NIH National Institute of General Medical Sciences (GM103310). We acknowledge the Weill Cornell Medicine Cryo-EM Core Facility for use of equipment, technical assistance, and expertise. Our work was supported in part by the resources and expert staff of Weill Cornell Medicine's Scientific Computing Unit. Molecular graphics were made and analysis was performed with UCSF ChimeraX, developed by the Resource for Biocomputing, Visualization, and Informatics at the University of California, San Francisco, with support from National Institutes of Health R01-GM129325 and the Office of Cyber Infrastructure and Computational Biology, National Institute of Allergy and Infectious Diseases. This work was funded by a Fonds de recherche du Québec – Santé (FRQS) Doctoral Fellowship to PMGEB, a Natural Sciences and Engineering Research Council of Canada (NSERC, PGS-D) doctoral award to AMP, a Leon Levy Fellowship in Neuroscience and a Weill Cornell Medical College startup grant to JRM, and Canadian Institutes of Health Research (CIHR) grants to DB (FRN 136832 and FRN 162317).

## Additional information

### Funding

| Funder | Grant reference number | Author |
|---|---|---|
| Leon Levy Foundation | | Joel R Meyerson |
| Fonds de Recherche du Québec - Santé | | Patricia MGE Brown |
| Canadian Institutes of Health Research | 136832 | Derek Bowie |
| Canadian Institutes of Health Research | 162317 | Derek Bowie |
| Natural Sciences and Engineering Research Council of Canada | | Amanda M Perozzo |
| Leon Levy Foundation | | Joel R Meyerson |

The funders had no role in study design, data collection and interpretation, or the decision to submit the work for publication.

### Author contributions

Nandish Khanra, Conceptualization, Data curation, Formal analysis, Validation, Investigation, Visualization, Methodology, Writing - original draft, Project administration, Writing - review and editing; Patricia MGE Brown, Amanda M Perozzo, Conceptualization, Data curation, Formal analysis, Validation, Investigation, Visualization, Methodology, Writing - review and editing; Derek Bowie, Conceptualization, Data curation, Formal analysis, Supervision, Funding acquisition, Validation, Investigation, Visualization, Methodology, Project administration, Writing - review and editing; Joel R Meyerson, Conceptualization, Data curation, Formal analysis, Supervision, Funding acquisition, Validation, Investigation, Visualization, Methodology, Writing - original draft, Project administration, Writing - review and editing

### Author ORCIDs

Nandish Khanra (iD) https://orcid.org/0000-0003-4217-1273
Patricia MGE Brown (iD) https://orcid.org/0000-0001-8340-0330
Amanda M Perozzo (iD) http://orcid.org/0000-0001-9681-3548
Derek Bowie (iD) http://orcid.org/0000-0001-9491-8768
Joel R Meyerson (iD) https://orcid.org/0000-0002-6127-0093

### Decision letter and Author response

Decision letter https://doi.org/10.7554/eLife.66097.sa1
Author response https://doi.org/10.7554/eLife.66097.sa2

## Additional files

### Supplementary files

- Transparent reporting form

### Data availability

Cryo-EM density maps have been deposited in the Electron Microscopy Data Bank (EMDB) under accession numbers EMD-23017 (GluK2/K5-apo), EMD-23014 (GluK2/K5-CNQX), and EMD-23015 (GluK2/K5-L-Glu). Model coordinates have been deposited in the Protein Data Bank (PDB) under accession numbers 7KS0 (GluK2/K5-CNQX) and 7KS3 (GluK2/K5-L-Glu). Raw cryo-EM data will be publicly available on the EMPIAR repository upon publication under the accession numbers: EMPIAR-10658, EMPIAR-10659, EMPIAR-10660.

The following datasets were generated:

| Author(s) | Year | Dataset title | Dataset URL | Database and Identifier |
|---|---|---|---|---|
| Khanra N, Brown PMGE, Perozzo AM, Bowie D, Meyerson JR | 2021 | Glu2/K5 apo | https://www.ebi.ac.uk/pdbe/emdb/empiar/entry/10658/ | Electron Microscopy Public Image Archive, EMPIAR-10658 |
| Khanra N, Brown PMGE, Perozzo AM, Bowie D, Meyerson JR | 2021 | GluK2/K5 with 6-Cyano-7-nitroquinoxaline-2,3-dione (CNQX) | https://www.ebi.ac.uk/pdbe/emdb/empiar/entry/10659/ | Electron Microscopy Public Image Archive, EMPIAR-10659 |
| Khanra N, Brown PMGE, Perozzo AM, Bowie D, Meyerson JR | 2021 | GluK2/K5 with L-Glu | https://www.ebi.ac.uk/pdbe/emdb/empiar/entry/10660/ | Electron Microscopy Public Image Archive, EMPIAR-10660 |
| Khanra N, Brown PMGE, Perozzo AM, Bowie D, Meyerson JR | 2021 | GluK2/K5 apo | http://www.ebi.ac.uk/pdbe/entry/emdb/EMD-23017 | Electron Microscopy Data Bank, EMD-23017 |
| Khanra N, Brown PMGE, Perozzo AM, Bowie D, Meyerson JR | 2021 | GluK2/K5 with 6-Cyano-7-nitroquinoxaline-2,3-dione (CNQX) | http://www.ebi.ac.uk/pdbe/entry/emdb/EMD-23014 | Electron Microscopy Data Bank, EMD-23014 |
| Khanra N, Brown PMGE, Perozzo AM, Bowie D, Meyerson JR | 2021 | GluK2/K5 with L-Glu | http://www.ebi.ac.uk/pdbe/entry/emdb/EMD-23015 | Electron Microscopy Data Bank, EMD-23015 |
| Khanra N, Brown PMGE, Perozzo AM, Bowie D, Meyerson JR | 2021 | GluK2/K5 with 6-Cyano-7-nitroquinoxaline-2,3-dione (CNQX) | https://www.rcsb.org/structure/7KS0 | RCSB Protein Data Bank, 7KS0 |
| Khanra N, Brown PMGE, Perozzo AM, Bowie D, Meyerson JR | 2021 | GluK2/K5 with L-Glu | https://www.rcsb.org/structure/7KS3 | RCSB Protein Data Bank, 7KS3 |

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
