## [Decision Letter]

**Acceptance summary:**

This work presents cryo-EM structures of heteromeric GluK2/K5 kainate receptors, which are ion channels that respond to the neurotransmitter glutamate. This is an important advance because previous structures had all been of homomeric receptors, while it is known that native receptors are largely heteromeric. This paper will be of great interest to molecular neuroscientists. It presents high-quality structural data of a critical synaptic ionotropic glutamate receptor in the context of its known functional properties.

**Decision letter after peer review:**

Thank you for submitting your article "Architecture and structural dynamics of the heteromeric GluK2/K5 kainate receptor" for consideration by *eLife*. Your article has been reviewed by four peer reviewers, including Merritt Maduke as the Reviewing Editor and Reviewer #1, and the evaluation has been overseen by Kenton Swartz as the Senior Editor. The following individual involved in review of your submission has agreed to reveal their identity: Andrew J R Plested (Reviewer #2).

The reviewers have discussed the reviews with one another and the Reviewing Editor has drafted this decision to help you prepare a revised submission.

Summary:

In this work, the authors solve structures of a kainate-type glutamate receptor in a hetermeric form that is expected to be a major form in the brain. By using antagonists and agonists, they could solve the structures in resting, inhibited and desensitized forms. The major strengths are the establishment of a clear assembly rule – the GluK5 subunits sit on the A/C positions – so called proximal subunits – and the GluK2 subunits sit in the B/D positions. Given the different functional properties of heteromeric receptors, it is a shock that the GluK2/K5 heteromers here present very similar structures to GluK2 homomers. These similarities suggest common mechanisms despite molecular diversity. The weakness of the work is that there is no independent test of the subunit arrangement, based on the structural knowledge delivered. Also, the low resolution of some parts of the structures forbids analysis of some aspects of the biology, including one of the linkers. This work will open the way to a more accurate molecular analysis of kainate receptors, and a better understanding of the diversity in the glutamate receptor family. Additional discussion as to how these structures provide insight into extensive functional data available for the heteromeric receptors will increase the impact of the paper.

Essential revisions:

1) It appears that the authors needed to remove particles that were not amenable to 3D reconstruction in their single particle analyses. After cleaning up the particles, were the authors able to observe meaningful conformational variations that may provide insights into functions? Or is there only one conformation of each for the CNQX-bound and the glutamate -bound (desensitized) structures? The authors may pick these conformational variations by running other software packages such as RELION and/or cisTEM. It is likely advantageous for the authors to run further 3D classification on their dataset to potentially extract meaningful 3D classes to enrich their insights into protein dynamics.

2) One major concern is that since these structures are with modified constructs and may not represent the predominant physiologically conformation. Hence it is essential to have other measurements, such as cross-linking/ mutational studies/or other biochemical/biophysical methods that can validate these structures. At a minimum the authors should discuss this and in addition rationalize the structure with prior functional data, adding the caveat that the subunit arrangement seen in the structures will be confirmed or supported by future experiments using an orthogonal technique (such as crosslinking).

3) Related to point 2: while it is good to see that the constructs used for cryo-EM constructs exhibit the expected electrophysiological properties (deactivation and densensitization), the functional characteristics are not directly compared to the wild-type constructs. It would be helpful to know whether the changes to the constructs introduce changes in receptor kinetics. If the authors do not have these data, the issue should at least be raised and discussed.

4) Additional discussion is needed as to how these structures provide insight into extensive functional data available for the heteromeric receptors. For example, DOI: 10.1016/j.bpj.2017.08.047 and other prior functional studies by co-author Bowie and by others (Barberis/Mulle for example). We found it strange such discussion was missing. (Some studies were cited, but the impression is that the structure adds little. We think the authors could either speculate or say what else has to be done before the new structures can be used to gain additional insight into mechanism; for example, answering questions such as why GluK2/5 have slower deactivation speed and faster desensitization speed compared to GluK2?).

5) That the structures of GluK5 and GluK2 are so similar that they must be distinguished by differential glycans is somewhat of a surprise. The authors do not comment, except the subunits are described as "mismatched". The phrase "mismatched subunits are woven together" is pretty, but is it apt? The subunits seem very similar…

6) Because particles are selected, can we be sure that the rule of assembly the authors propose (GluK5 is always at A/C) is really validated? Are there any further information that they authors can provide to further assure this proposal? Or perhaps to add a caveat?

7) Subsection “Receptor isolation, functional testing, and structure determination”: The “tilt” is not clear to readers. A figure to explain this point is necessary. Or modify Figure 1—figure supplement 4 to address this point. Also, it is not clear why the authors think this tilt is the reason for not observing cryo-EM density for the linker between ATD and LBD. How is that important in terms of functions of GluK2/5?

8) Subsection “Receptor organization and symmetry”. While CNQX density is not observed, can the authors say something about the LBD conformations? For example, are the LBD conformations similar to structure of antagonist-bound LBD crystal structures? Usage of "a saturating concentration of CNQX" may be a weak argument for stating this structure as CNQX-bound.

9) Discussion: We do not see a “pinwheel-shaped” motif in Figure 4. Do the authors mean Figure 5? Also, how exactly is this motif defined (e.g. what residue, where in the structure, etc?)

10) Can the authors gain any insights into the mechanism underlying channel regulation by binding of extracellular ions from the cryo-EM structures?

11) Is there anything new gained from the 2/5 ATD tetramer considering that the ATD layer has the highest resolution in both CNQX and glutamate structures? Is there any opportunity for targeting ATD as in the case of NMDA receptors (e.g. ifenprodil binding to the NMDAR ATD)? This could be addressed in the manuscript.

Queries about writing/interpretation:

12) Abstract: "The receptor assembles with two copies of each subunit and positions GluK5 subunits proximal to the channel to direct receptor activation. "

We are not sure that we agree with the logic of this statement. It is clear from previous work that B/D subunits provide (if anything) more driving force to open the channel. But then the authors do not follow up on this in the main text so perhaps it should be removed. How does K5 "direct" activation?

13) "Critically, this work gives a foundation for understanding differences and similarities among KAR, AMPAR (Herguedas et al., 2019; Zhao et al., 2019), and NMDAR structures (Karakas and Furukawa, 2014; Lee et al., 2014; Lü et al., 2017), and reveals that while they share common architectural principles the iGluR receptor subtypes display enormous structural diversity."

The point about structural diversity is well taken, but the new knowledge contributed by this paper does not go in this direction. This would be in comparison to the cited works and the senior author's own previous work on KARs, which oddly merits no citation. We do not want to diminish the achievement in this paper. That GluK2/K5 seem to behave almost the same as GluK2 homomer is really interesting, but it does not expand our view of structural diversity. Finally, it would be appropriate to mention GluD structures from Janesh Kumar's lab if you are talking about diversity in the family.

14) Subsection “Receptor isolation, functional testing, and structure determination”: The structure in the GluA2 paper (2009) is the antagonist-bound state, not the resting state.

15) "Furthermore, close inspection shows GluK5em G helices reside ~3 Å closer to each other across the two-fold symmetry plane of the LBD layer, compared to analogous GluK2 subunits in the homomer (Figure 4E, H). This subtle difference has a geometric consequence of producing a more compact desensitization ring in the heteromer (Figure 4E, H, 226 insets)."

Does not go with the next sentence "This discovery strongly suggests that despite different receptor compositions KAR homomers and heteromers employ a shared desensitization mechanism."

16) "the GluK2/K5em structure raises the question of whether KAR tri-heteromers could form". We do not understand the logic of this point. Of course it's an interesting question – heteromeric receptors have been studied for decades and this possibility is mentioned at the end of the Introduction of the Cui and Mayer, 1999. How is the question raised by the work in hand? The idea is that a third subunit like GluK1 would have to occupy a B or D position because GluK5 is dominating assembly? As we know that K1/K2 heteromers and homomers can form, we see no restriction on which position that GluK1 would adopt if it were to be introduced to the GluK2/5 heteromer. If there is a more profound observation that we are missing, the authors should try to illustrate it more clearly, perhaps with an additional cartoon.

17) "This study marks a milestone in iGluR biology by providing the first structures of the major neuronal KAR to accompany structures of the dominant forms of NMDARs and AMPARs."

We think it is wrong to say "neuronal KAR" because now we have (and there will be more) structures of native receptors. These GluK2/K5 are the presumed major neuronal type. It is good to have KAR heteromer structures, but we would not be so effusive. For example, GluK2 was not edited at Q/R in this study; in the forebrain, it would be (90%, Bernard et al. 1999 EJN).

---

## [Author Response]

Essential revisions:1) It appears that the authors needed to remove particles that were not amenable to 3D reconstruction in their single particle analyses. After cleaning up the particles, were the authors able to observe meaningful conformational variations that may provide insights into functions? Or is there only one conformation of each for the CNQX-bound and the glutamate -bound (desensitized) structures? The authors may pick these conformational variations by running other software packages such as RELION and/or cisTEM. It is likely advantageous for the authors to run further 3D classification on their dataset to potentially extract meaningful 3D classes to enrich their insights into protein dynamics.

We share the reviewer’s interest in this question and looked closely for interpretable classes that could give meaningful insights into conformational variation. At each iteration of three-dimensional classification the class averages were carefully compared to one another. The classes typically had a range of resolutions and feature qualities as seen in our supplementary figures (Figure 1—figure supplement 4, Figure 1—figure supplement 6, and Figure 4—figure supplement 1), so we looked primarily for evidence of major differences in LBD or ATD domain orientations. Although we processed the data multiple times and used classification tools in both Relion and cryoSPARC we were unable to extract classes in obviously distinct conformations on the path to resolving full-length structures.

However, as described in the manuscript, after producing full-length reconstructions we then refined the ATD and LBD-TMD assemblies separately for both the GluK2/K5_em_-CNQX and GluK2/K5_em_-L-Glu structures. The fact that reconstructions of isolated ATD and LBD-TMD regions were better resolved than the full-length receptors suggests some degree of relative conformational mobility in the ATD versus the LBD-TMD assembly.

This also raised the question of local conformational variability within the ATD, LBD, and TMD regions. The ATDs were resolved to high resolution and we found no evidence for additional conformations during classification. However, the lower resolution of the LBD-TMD assemblies suggested local conformational variability that may have resisted classification. We attempted to refine the LBD and TMD layers separately but were unsuccessful, likely because of the small masses of these regions (120 kDa for the LBD layer; 60 kDa for the TMD layer). This motivated us to apply conformational variability analysis (cryoSPARC) to the LBD-TMD assemblies. Unfortunately this analysis showed no evidence for local conformational variability in either the LBD or the TMD.

In summary, we found one conformation for each full-length structure. Local reconstructions of the ATD and LBD-TMD assemblies indicated that these two regions are mobile with respect to one another. Further investigation of local variability within the ATD, LBD, and TMD regions showed no evidence of meaningful conformational variation within these regions of the structures.

We feel this is an important point to address and have elaborated on our approach in the Materials and methods section (see Materials and methods section, “Image processing and structure analysis” subsection, paragraph five).

2) One major concern is that since these structures are with modified constructs and may not represent the predominant physiologically conformation. Hence it is essential to have other measurements, such as cross-linking/ mutational studies/or other biochemical/biophysical methods that can validate these structures. At a minimum the authors should discuss this and in addition rationalize the structure with prior functional data, adding the caveat that the subunit arrangement seen in the structures will be confirmed or supported by future experiments using an orthogonal technique (such as crosslinking).

We agree that applying an additional technique would add value by further validating our expression constructs and structures. Because the present restrictions on lab operations pose a challenge to timely preparation of new data, the subunit arrangement seen in the structures will be confirmed or supported by future experiments using an orthogonal technique such as crosslinking. We have noted this in the text to inform the reader (see Discussion, paragraph seven).

3) Related to point 2: while it is good to see that the constructs used for cryo-EM constructs exhibit the expected electrophysiological properties (deactivation and densensitization), the functional characteristics are not directly compared to the wild-type constructs. It would be helpful to know whether the changes to the constructs introduce changes in receptor kinetics. If the authors do not have these data, the issue should at least be raised and discussed.

Thank you for this comment. Our initial submission did indeed compare wild-type to cryo-EM constructs. This was presented in the Results section and the data were found in Figure 1—figure supplement 2. Specifically, wild-type function was shown in panels B-D, and cryo-EM construct function in panels E-G. In addition, Table 1 provided a summary of all the kinetic properties (deactivation, desensitization time constants) and reveals that the functional behavior of the wild-type and cryo-EM KAR heteromers are very similar.

This reviewer comment prompted us to re-evaluate our presentation of the functional data and we felt it was essential to improve communication of these data in the text. Accordingly we have revised the Results subsection dealing with functional validation (see Results), and we more clearly alert the reader to Figure 1—figure supplement 2 and Table 1.

4) Additional discussion is needed as to how these structures provide insight into extensive functional data available for the heteromeric receptors. For example, DOI: 10.1016/j.bpj.2017.08.047 and other prior functional studies by co-author Bowie and by others (Barberis/Mulle for example). We found it strange such discussion was missing. (Some studies were cited, but the impression is that the structure adds little. We think the authors could either speculate or say what else has to be done before the new structures can be used to gain additional insight into mechanism; for example, answering questions such as why GluK2/5 have slower deactivation speed and faster desensitization speed compared to GluK2?).

We have expanded the Discussion of how the structures in this study relate to functional data on desensitization, deactivation, and to emerging data that suggest the ion-dependent gating of homomeric KARs differs from the GluK2/K5 heteromer. We also comment on the future prospect of using the experimental foundation established in this study to address questions about ion binding, ligand binding, and receptor activation (see Discussion).

5) That the structures of GluK5 and GluK2 are so similar that they must be distinguished by differential glycans is somewhat of a surprise. The authors do not comment, except the subunits are described as "mismatched". The phrase "mismatched subunits are woven together" is pretty, but is it apt? The subunits seem very similar…

In order to improve clarity of the text we changed the phrase “how its mismatched subunits are woven together” to “how its subunits are organized” (see Abstract).

6) Because particles are selected, can we be sure that the rule of assembly the authors propose (GluK5 is always at A/C) is really validated? Are there any further information that they authors can provide to further assure this proposal? Or perhaps to add a caveat?

We were also very interested in this question and analyzed the data multiple times throughout the study in the hopes of discovering receptors that did not obey the proposed assembly. Despite extensive three-dimensional classification we never found evidence that such particle subsets exist in the datasets. Given the ten-fold excess of GluK5_em_ versus GluK2_em_ virus added during protein expression, we would expect GluK5 to adopt alternative positions (i.e. B or D) if this were possible. That this does not happen led us to conclude that GluK5 is indeed always at the A/C positions. We have now addressed this point in the Discussion section (see Discussion, paragraph six).

7) Subsection “Receptor isolation, functional testing, and structure determination”: The “tilt” is not clear to readers. A figure to explain this point is necessary. Or modify Figure 1—figure supplement 4 to address this point. Also, it is not clear why the authors think this tilt is the reason for not observing cryo-EM density for the linker between ATD and LBD. How is that important in terms of functions of GluK2/5?

We have added annotations to Figure 1—figure supplement 4 in order to indicate the ATD “tilt”.

We did not intend to imply or argue that the ATD tilt is the reason for not observing cryo-EM density in the ATD-LBD linkers. Rather, we merely meant to say that the tilted ATD layer, together with the poorly resolved linkers, may indicate flexibility between the ATD layer and the rest of the receptor. Indeed, resolution improvements achieved through local refinement of the ATD region and LBD-TMD region in the GluK2/K5_em_-CNQX structure support this conclusion (see our response to Comment 1 above). Because similar features have been observed in AMPAR structures using both X-ray crystallography (Sobolevsky, Rosconi and Gouaux, 2009) and cryo-EM (Nakagawa, 2019) the “tilt” may be some basic feature of KARs and AMPARs structures that is not yet understood. We have revised the text to clarify this point (see Results).

8) Subsection “Receptor organization and symmetry”. While CNQX density is not observed, can the authors say something about the LBD conformations? For example, are the LBD conformations similar to structure of antagonist-bound LBD crystal structures? Usage of "a saturating concentration of CNQX" may be a weak argument for stating this structure as CNQX-bound.

This is an absolutely essential point to address. In our initial submission we did indeed compare LBDs from GluK2/K5_em_-CNQX to crystal structures of isolated LBDs. The analysis can be found in Figure 1—figure supplement 5B and shows that the LBDs from GluK2/K5_em_-CNQX have similar conformations to the antagonist-bound LBD reference structure. In addition, we showed analogous comparisons for GluK2/K5_em_-apo and GluK2/K5_em_-L-Glu which also support our conclusions about the LBD conformations for those structures.

We believe this analysis was not communicated clearly in the initial submission. To address the deficiency and expand the analysis we have done the following:

a) Added text in the manuscript to better alert the reader to the analysis (see Results).

b) Added RMSD values to quantify how closely the GluK2_em_ and GluK5_em_ LBDs match the reference LBDs used for comparison (see Figure 1—figure supplement 5).

c) Revised the layout of Figure 1—figure supplement 5 to improve the presentation.

9) Discussion: We do not see a “pinwheel-shaped” motif in Figure 4. Do the authors mean Figure 5? Also, how exactly is this motif defined (e.g. what residue, where in the structure, etc?)

The “pinwheel” referred to the entire LBD layer in the GluK2/K5-L-Glu structure when viewed from the extracellular side (Figure 4C and D). It was so named because we felt it resembled a classic pinwheel toy and the naming might aid reader interpretation of the structure. However, we now appreciate it may not appear this way to every reader. As such we have removed the “pinwheel” language from the manuscript and replaced it with plain language descriptions.

10) Can the authors gain any insights into the mechanism underlying channel regulation by binding of extracellular ions from the cryo-EM structures?

The resolution in the LBDs was not high enough to visualize ions so we were unable to gain insights into channel regulation by extracellular ion binding. However, this is certainly an area for future investigation and we have commented on this point in the Discussion (see Discussion, paragraph four).

11) Is there anything new gained from the 2/5 ATD tetramer considering that the ATD layer has the highest resolution in both CNQX and glutamate structures? Is there any opportunity for targeting ATD as in the case of NMDA receptors (e.g. ifenprodil binding to the NMDAR ATD)? This could be addressed in the manuscript.

In our analysis we found that the ATDs in CNQX and L-Glu structures do not show conformational differences. This is consistent with the idea that KAR ATDs are not involved in channel gating. Rather, they are thought to play a key role in subunit oligomerization during receptor biogenesis. We have added text to the Discussion section to better contextualize the ATD results (see Discussion, paragraph five). It is currently unclear if KAR ATDs participate in small molecule binding and thus appear to be different from NMDAR ATDs in this respect.

Queries about writing/interpretation:12) Abstract: "The receptor assembles with two copies of each subunit and positions GluK5 subunits proximal to the channel to direct receptor activation. "We are not sure that we agree with the logic of this statement. It is clear from previous work that B/D subunits provide (if anything) more driving force to open the channel. But then the authors do not follow up on this in the main text so perhaps it should be removed. How does K5 "direct" activation?

We used the word “direct” in an effort to succinctly connect our results to the idea that GluK5 alone can activate the channel. However, after considering the reviewer feedback on this point we feel this language detracts from the clarity of the Abstract. We have replaced this wording with text to clearly state key findings from our structural analysis (see Abstract).

13) "Critically, this work gives a foundation for understanding differences and similarities among KAR, AMPAR (Herguedas et al., 2019; Zhao et al., 2019), and NMDAR structures (Karakas and Furukawa, 2014; Lee et al., 2014; Lü et al., 2017), and reveals that while they share common architectural principles the iGluR receptor subtypes display enormous structural diversity."The point about structural diversity is well taken, but the new knowledge contributed by this paper does not go in this direction. This would be in comparison to the cited works and the senior author's own previous work on KARs, which oddly merits no citation. We do not want to diminish the achievement in this paper. That GluK2/K5 seem to behave almost the same as GluK2 homomer is really interesting, but it does not expand our view of structural diversity. Finally, it would be appropriate to mention GluD structures from Janesh Kumar's lab if you are talking about diversity in the family.

We thank the reviewer(s) for making this point and have removed the language on structural diversity from the sentence (see Introduction, paragraph four). The addition of references to earlier KAR structures and GluD structures is indeed appropriate and we have also added these references.

14) Subsection “Receptor isolation, functional testing, and structure determination”: The structure in the GluA2 paper (2009) is the antagonist-bound state, not the resting state.

We have now changed this reference to Dürr et al., 2014 which reports an apo GluA2 structure (PDB: 4U2P) (see Results).

15) "Furthermore, close inspection shows GluK5em G helices reside ~3 Å closer to each other across the two-fold symmetry plane of the LBD layer, compared to analogous GluK2 subunits in the homomer (Figure 4E, H). This subtle difference has a geometric consequence of producing a more compact desensitization ring in the heteromer (Figure 4E, H, 226 insets)."Does not go with the next sentence "This discovery strongly suggests that despite different receptor compositions KAR homomers and heteromers employ a shared desensitization mechanism."

Thank you, we have removed this sentence (see Results).

16) "the GluK2/K5em structure raises the question of whether KAR tri-heteromers could form". We do not understand the logic of this point. Of course it's an interesting question – heteromeric receptors have been studied for decades and this possibility is mentioned at the end of the Introduction of the Cui and Mayer, 1999. How is the question raised by the work in hand? The idea is that a third subunit like GluK1 would have to occupy a B or D position because GluK5 is dominating assembly? As we know that K1/K2 heteromers and homomers can form, we see no restriction on which position that GluK1 would adopt if it were to be introduced to the GluK2/5 heteromer. If there is a more profound observation that we are missing, the authors should try to illustrate it more clearly, perhaps with an additional cartoon.

In this paragraph we merely wished to make the point that we find no obvious aspects of the GluK2/K5 subunit arrangement which would preclude formation of KAR tri-heteromers containing GluK2, GluK5 and another subunit. To our knowledge KARs have so far been studied as homomers or di-heteromers so we thought it would be stimulating to the reader to address the prospect of more complex KARs. Indeed, we felt this merited discussion given the recent structural evidence for native AMPAR tri-heteromers from the Gouaux lab, and well-known existence of NMDAR tri-heteromers. However, upon considering reviewer feedback we believe that this point is not strong enough to warrant discussion and as such have removed the paragraph to avoid confusion.

17) "This study marks a milestone in iGluR biology by providing the first structures of the major neuronal KAR to accompany structures of the dominant forms of NMDARs and AMPARs."We think it is wrong to say "neuronal KAR" because now we have (and there will be more) structures of native receptors. These GluK2/K5 are the presumed major neuronal type. It is good to have KAR heteromer structures, but we would not be so effusive. For example, GluK2 was not edited at Q/R in this study; in the forebrain, it would be (90%, Bernard et al. 1999 EJN).

As suggested, we have replaced “major neuronal” with “presumed major neuronal type” (see Discussion, paragraph eight).